# Climatology of Large Hail in Europe: Characteristics of the European Severe Weather Database

Faye Hulton[1,a], David M. Schultz[1,2]

[1] Centre for Atmospheric Science, Department of Earth and Environmental Sciences, University of Manchester, Manchester, M13 9PL, United Kingdom

[2] Centre for Crisis Studies and Mitigation, University of Manchester, Manchester, M13 9PL, United Kingdom

[a] Now at: MetDesk, Aylesbury, HP22 6NJ, United Kingdom

*Correspondence to*: Faye Hulton, faye.hulton@gmail.com

Submitted as an Article to *Natural Hazards and Earth System Science,* 4 February 2023, Revised 12 February 2024

**Abstract.** Large hail (greater than 2 cm in diameter) can cause devastating damage to crops and property, and can even cause loss of life. Because hail reports are often collected by individual countries, constructing a European-wide large-hail climatology has been challenging to date. However, the European Severe Storm Laboratory's European Severe Weather Database provides the only pan-European dataset for severe convective storm reports. The database is comprised of 62,053 large-hail reports from 40 C.E. to September 2020, yet its characteristics have not been evaluated. Thus, the purpose of this study is to evaluate hail reports from this database for the purposes of constructing a climatology of large hail. For the period 2000–2020, large-hail reports are most prominent in June, whereas large-hail days are most common in July. Large hail is mostly reported between 1300–1900 local time, a consistent pattern since 2010. The intensity, as measured by maximum hail size, shows decreasing frequency with increasing hailstone diameter, and little change over the 20-year period. The quality of reports by country varies, with the most complete reporting being from central European countries. These results suggest that despite its short record, many indications are that the dataset represents some reliable aspects of European large-hail climatology, albeit with some limitations.

## 1 Introduction

Hail with a diameter of at least 2 cm in the longest direction is called *large hail*, and it can cause damage to crops, property, or even loss of life. Several recent studies have documented the occurrence and variability of large hail, with special emphasis on the United States and Europe where large hail is common (e.g., Allen and Tippett 2015; Punge and Kunz 2016; Brooks et al. 2019; Púčik et al. 2019; Tang et al. 2019; Taszarek et al. 2020; Raupach et al. 2021). The strongest severe convective storms in Europe are often perceived to be less intense than the strongest storms in the United States, although they can be just as damaging. For example, one of the most devastating large-hail events took place over Germany on 28 July 2013 when two supercells formed almost simultaneously, producing hailstones of up to 10 cm in diameter and more than EUR 1 billion in insurance payouts (Kunz et al. 2018). Other similar events occurred over southern Germany on 10–12 June 2019, with one storm producing 6-cm hailstones and causing EUR 1 billion in damages (Wilhelm et al. 2021). More recently, several large-hail events were reported during summer 2021 in Poland, the Czech Republic, Germany, and Italy, with reported maximum hail sizes in excess of 7 cm (Associated Press 2021; Space 2021a,b,c). Although these extreme events are widely reported by the media, meteorological research on these storms may be hindered by the lack of ground-truth hail data, such as onset and ending times, duration, and hailstone size.

Such hail data in Europe is generally collected on a national scale, and hence most climatologies are produced
on a country-by-country basis (e.g., Brooks et al. 2009). Given the relatively small sizes of many European
countries, each country has a low probability of large hail occurring at any given time (e.g., Brooks et al. 2019).
A summary table of past European hail climatologies can be found in Tuovinen et al. (2009), and an updated
review was published by Punge and Kunz (2016). Because countries that have a similar spatial extent as Europe
have produced their own climatologies—such as the United States (Tang et al. 2019), Canada (Etkin and Brun
2001), and China (Zhang et al. 2008)—a pan-European large-hail climatology would be highly desired.
Climatologies of European convective storms and their impacts have been constructed using a number of
datasets.  For example, some studies have examined the climatology of convective storms using remote-sensed
data such as lightning, radar, and satellite (e.g., Punge et al. 2017).  Others have examined the environments that
favor such storms, such as through reanalyses or soundings (Rädler et al. 2018; Taszarek et al. 2017, 2018, 2019)
or reanalyses coupled with hailpad data (Sanchez et al. 2017).
To create a pan-European dataset of in situ surface reports from severe convective storms (including large
hail, tornadoes, and severe wind gusts), the European Severe Storms Laboratory formed the European Severe
Weather Database (ESWD) in 2006 (Dotzek et al. 2009; Groenemeijer et al. 2017). In addition to collecting
contemporary data, the ESWD has an ongoing objective of synthesizing historical large-hail data which helps
produce a longer and more complete climatology. Despite the tremendous potential value of the ESWD being the
only pan-European large-hail dataset, its characteristics need to be examined to understand its suitability for
answering certain scientific questions about large hail.  For example, Taszarek et al. (2019) found substantial
variability across Europe in the frequency of ESWD reports and the frequency of favorable environments for
convective storms.
To this effect, Púčik et al. (2019) constructed a climatology of large hail from the ESWD.  They examined
hail size, occurrence, annual cycle, diurnal cycle, and societal impacts (e.g., damages, injuries) for 39,537 reports
during the 13-yr period 2006–2018.  Although their work shed the first light on the pan-European distribution and
characteristics of large hail and large-hail days from surface reports, they concluded by foreseeing "an update to
this study as the reporting homogeneity improves in future."  In the present article, we explore whether increasing
the size of the dataset through lowering the quality-control levels of the reports and extending the period of
analysis yields comparable results, increasing the generality of Púčik et al.'s (2019) results.  In doing so, we also
document the reporting characteristics of the database as a function of time both throughout the 20th century and
within the last 20 years. In particular, we seek the possible existence of a relatively homogeneous period of time
in the database that could be used as a baseline for climatologies and climate-change studies.
This article consists of nine sections. Section 2 describes the data from the ESWD used in the present study.
Section 3 discusses the frequency of large-hail reports and days on decadal, annual, and diurnal time scales.
Section 4 investigates the intensity distribution of large hail, as segregated into 1-cm diameter bins, and discusses
how the frequency of large-hail size has changed over the past 20 years. Section 5 looks at the time accuracy of
these reports, how it has changed over the past 20 years, and how it varies by individual countries. Section 6
investigates the spatial distribution of reports by country. Because of the large number of reports from Poland
during the 1930s to 1950s, section 7 focuses on the data from Poland, comparing the historical frequency of reports
during this period to that from the period 2000–2020. Section 8 offers a discussion comparing our work to previous
hail climatologies and reflects on the prospects of using the ESWD as a baseline for climate-change research.
Section 9 summarizes the findings of this paper.

**2 Data and methods**
The climatology of European large hail in this present article is produced from the ESWD (Dotzek et al. 2009;
Groenemeijer et al. 2017). Large hail in the ESWD is defined as hail with a diameter of at least 2 cm in the longest
direction (Groenemeijer and Liang 2020), comparable to the severe-hail criterion of 0.75 inch (1.9 cm) in the
United States. The current ESWD data on hail is a mixture of historical entries, insurance data information, reports
provided by storm-spotters, national European meteorological organizations, and public entries via the ESWD
website at www.eswd.eu (Dotzek et al. 2009). Since December 2015, reports have also been collected via ESSL's
European Weather Observer app (Groenemeijer et al. 2017).
At the time this study commenced, the ESWD consisted of 62,053 large-hail reports from 59 countries dating
from 40 C.E. to 26 September 2020. All reports with hail sizes less than 2 cm were removed. Of the 59 countries
included with the initial dataset received from the European Severe Storms Laboratory, only 41 were in Europe.
Of those removed, the highest reporting countries were Turkey, Armenia, and Azerbaijan. Reports from other
countries that were removed included Morocco, Turkmenistan, Egypt, and Jordan. The Russian Federation was
included in the present study, even though a small number of reports were from the Asian part of the country. A
small part of Turkey is geographically in Europe, but their data was not included in this study.
We also examined two periods of time from the ESWD. The first period is the nearly 121-yr period from 1
January 1900 to 26 September 2020 (when work on this research commenced). We hereafter refer to this period
as 1900–2020, recognizing the omission of data from the last three months and four days of 2020. The second
period is more focused on the most recent large-hail data for the nearly 21-yr period 1 January 2000 to 26
September 2020, hereafter referred to as 2000–2020.
All data is imputed in a standard format and is given a single quality-control level by the maintenance team
(Dotzek et al. 2009). There are four quality-control levels given to these entries (Groenemeijer and Kühne 2014):
• Q0: "as received", any report straight from the public,
• QC0+: "plausibility checked", any report checked by staff at the European Severe Storms Laboratory or a
partner organization,
• QC1: "report confirmed", any report confirmed by a reliable source such as a national meteorological
organization or storm-spotter network, and
• QC2: "event fully verified", any report from an event that has been subject of a scientific case study.
As mentioned in section 1, Púčik et al. (2019) used only QC1 and QC2 events. However, to see if the quality-
control level affects the interpretation of the results, this present study uses QC0+, QC1, and QC2. For the period
1900–2020, there were 9173 QC0+, 45,805 QC1, and 2391 QC2 reports, producing a total of 57,369 large-hail
reports. For the period 2000–2020, there were 6330 QC0+, 20,585 QC1, and 1310 QC2 reports, producing a total
of 28,225 large-hail reports. Thus, the addition of the QC0+ reports increased the size of the 1900–2020 dataset
by 19% and the 2000–2020 dataset by 29%.
With these two datasets constructed, we can then look at their characteristics. In particular, we are
interested in the number of large-hail days, size of the large-hail reports, and time accuracy of the reports. The
annual number of large-hail days was derived from the annual number of large-hail reports by removing duplicate
dates. We analyzed not only the number of hail reports, but the number of hail days, as well. Hail days are a more
robust measure of hail occurrence and helps minimize variability due to variability in hail reporting across
different countries. Hail days are also useful for certain purposes. For example, Punge and Kunz (2016) wrote that
the insurance industry measures hail damage per region per day instead of measuring damage per individual
hailstorm. Therefore, a pan-European overview of hail days may be of use given that these insurance portfolios
cover large parts of Europe, often including data from multiple countries. However, an awareness of the spatial
distribution of these reports is necessary to identify the most at-risk regions.
The size of the hail in each hail report was defined as the maximum hail diameter recorded in cm. Although
the ESWD contains fields for the fall speed and density of the hailstones, these were infrequently reported and
were not considered as part of the present article. To represent the size distribution of the reports, the reports were
classified into 1-cm bins based on their maximum hail diameter, starting at the minimum threshold of large hail
of 2 cm. The *time accuracy* of reports is a field in the ESWD that allows the user to know how reliable the
reporting time of the large-hail report is. The time accuracy represents the total time window that a given report
was recorded in. For example, a 30-min time accuracy would indicate that the hail fell in the window of 15 min
before the recorded time to a maximum of 15 min after the recorded time. The existing ESWD dataset is a result
of both meteorological variations in hail and reporting issues, much as other severe-weather datasets have (e.g.,
Groenemeijer and Kühne 2014; Punge and Kunz 2016; Antonescu et al. 2017; Púčik et al. 2019). Indeed,
underreporting from rural areas and nighttime storms may influence this dataset. These and other characteristics
of the large-hail dataset will be explored in subsequent sections.


## 3 Frequency of large hail across Europe: 1900–2020

To understand the number of large-hail reports as a function of time, the annual number of large-hail reports
and annual number of large-hail days were plotted versus year from 1900 to 2020 (Fig. 1). Throughout much of
this period, the annual number of reports was quite small, with peaks during the 1930s, 1940s–1950s, and early
1980s before a steady increase starting around 2000. These two peaks in the 1930s and 1940s–1950s were
associated with a large number of reports from Poland and are investigated further in section 8. The lesser peak
during the 1980s was associated with a number of reports from Italy, but is not considered further.
Figure 1 also shows the annual number of hail days from 1900 to 2020. The peaks in large-hail days during
the 1930s and 1940s–1950s suggest that there were many large-hail events, not just many reports. Moreover, these
periods illustrate that, while some periods and some locations may be well represented in the database, reporting
of large hail throughout much of the 20th century in the ESWD is far from complete.
Focusing on the last 30 years, the number of reports increased starting around 2000 and continued to rise until
2020. (Recall that the 2020 data was only available until 26 September, which may explain the fewer number
reports, although most large-hailfall in Europe is reported between April and September.) In contrast, the number
of large-hail days began rising a few years earlier in the late 1990s before reaching a plateau during the 2010s
with around 175 annual large-hail days per year, similar to Taszarek et al. (2020, their Fig. 2a). This result suggests
that the database grew around this time by first obtaining data from a larger number of days on which hail fell,
followed by the database growing with a larger number of reports within the same day. The inconsistency in
reports over time is also seen in other convective-storm research, such as for tornadoes as described by Antonescu

 et al. (2017), and may be a reflection in scientific interest in severe convective storms, or due to economic or

 political changes.

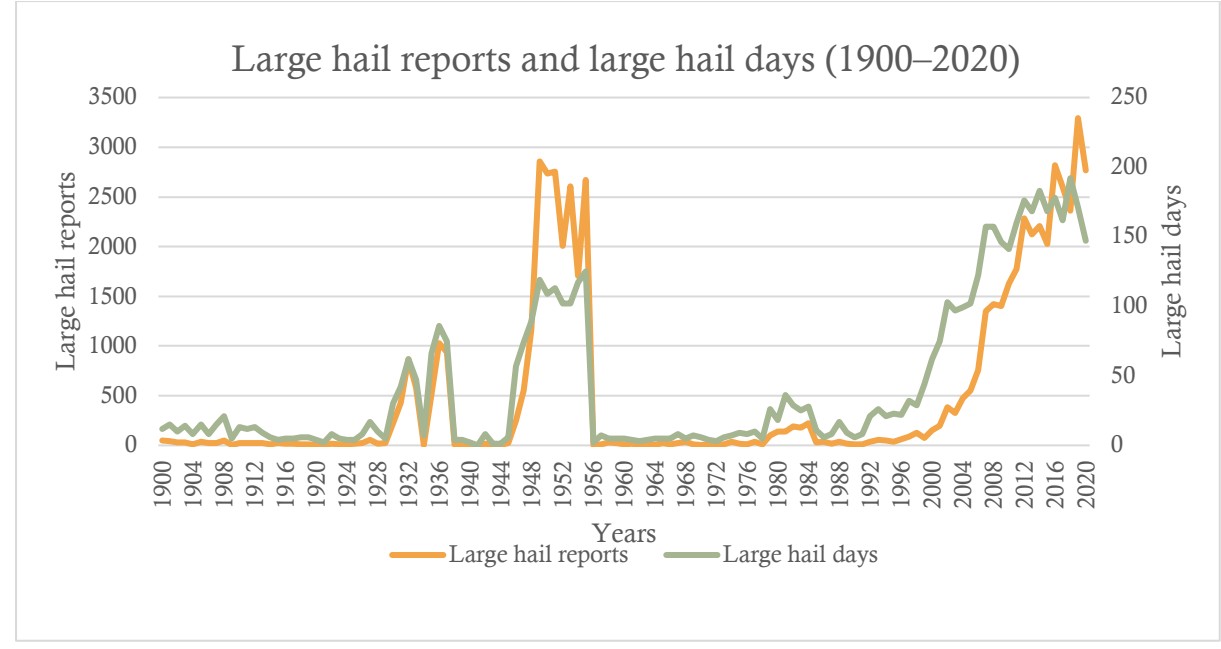

 **Figure 1. Time series of annual numbers of large-hail reports (orange line) and large-hail days (green**

 **line) across Europe 1900–2020.**

To show these data in a slightly different way, a scatterplot is created of the number of hail days versus
number of hail reports for each year in the dataset, with different colors for the period before and after 2000 (Fig.
2). The dataset from 1900 onwards suggests a positive linear relationship between large-hail reports and large-
hail days; however, the spread is sometimes large. The high number of large-hail reports during 1949–1955
(mostly from Poland, section 8) and early 1950s all congregate in one region of the graph and 2010–2020 also
congregate in one region. As fewer reports are needed for a greater quantity of large-hail days, either areal extent
of spotters has improved, the number of reporters has decreased in hail-prone regions, or the ESWD maintenance
team have improved their ability to detect reports linked to the same event. Thus, the 1950s are a time when
reports mostly came from Poland (section 8) and captured a large number of large-hail days, indicating that certain
periods of time can be fruitful for hail research using the ESWD. The spatial distribution of these reports is
discussed in section 7.


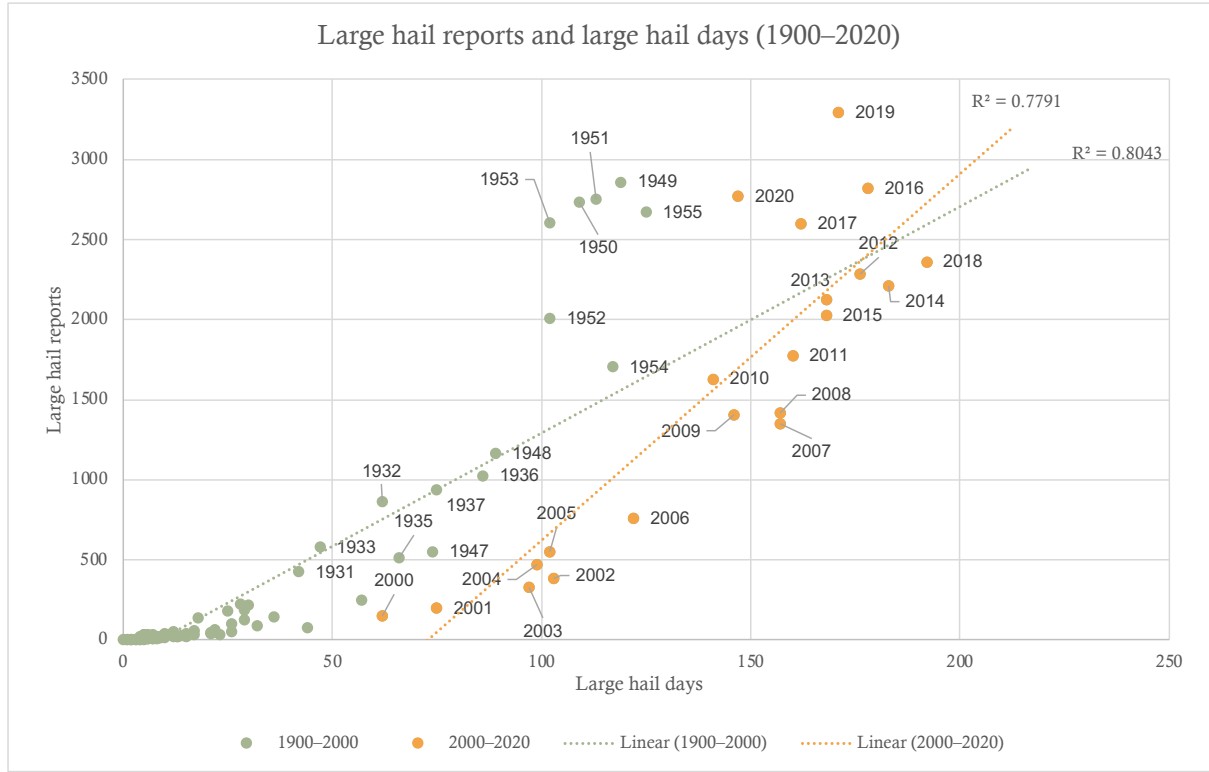


**Figure 2. Scatterplot of the annual number of large-hail days versus annual number of large-hail reports**
**across Europe: 1900–2000 (green dots) and 2000–2020 (orange dots), with corresponding linear regression**
**lines. These quantities are not divided by the number of years because of the incomplete data for the year**
**2020.**

The average monthly distribution of the number of large-hail reports and large-hail days from 2000 to 2020 is plotted in Fig. 3. The warm-season months of May, June, and July have the highest number of large-hail reports, and the cool-season months from October to March have the lowest. Whereas the month with the highest number of large-hail reports is June, the month with the highest number of large-hail days is July. Figure 3 can be compared to Púčik et al. (2019, their Fig. 4) who break down the annual cycle into the frequency of reports for the continental regions of Europe north of 46°N and the more Mediterranean-influenced regions south of 46°N. Despite these differences, these two distributions look similar, with the added information coming from the distribution of large-hail days in the present study. The distribution of large-hail days in Fig. 3 is more similar to the shape of the distribution of north of 46°N in Púčik et al. (2019, their Fig. 4), meaning that fewer reports occur later in the season although the number of large-hail days remains relatively high. These distributions are also similar to those from Kunz et al. (2020, their Fig. 2a) for hailstorms in central Europe using radar-derived hail streaks combined with all quality levels from the ESWD, indicating that this larger dataset including QC0+ events derived using different methods is a reliable source of large-hail data.

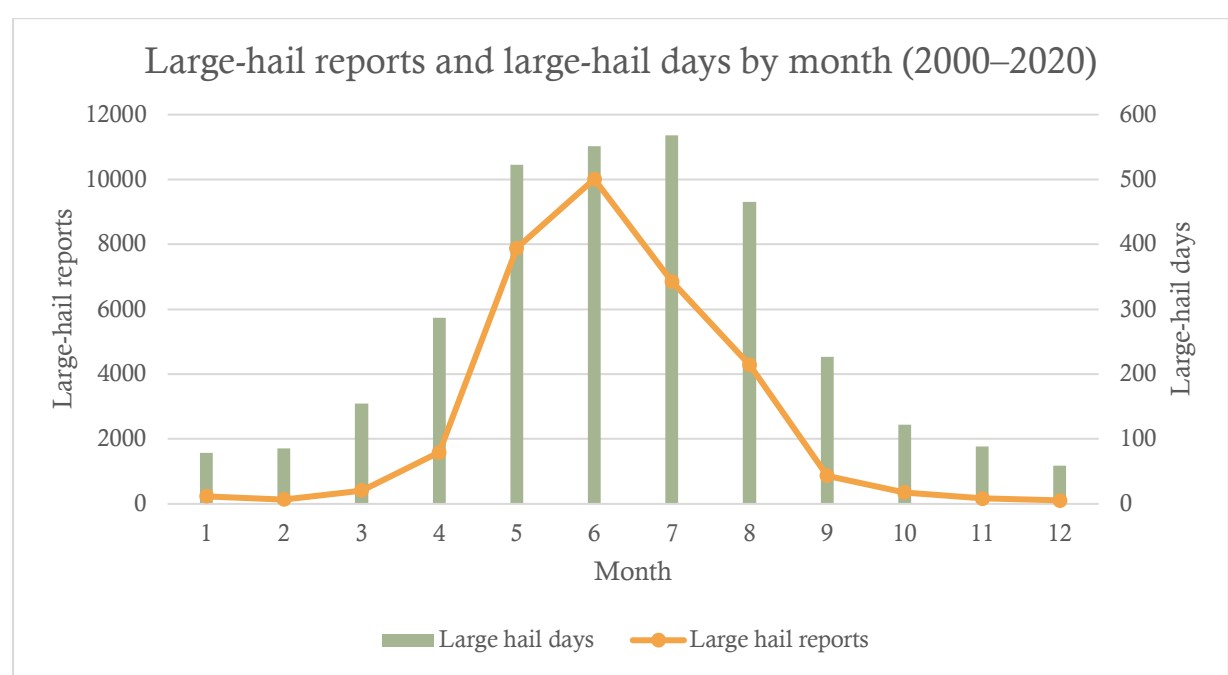

**Figure 3. Combined line graph and bar chart of the total monthly numbers of large-hail reports (orange line) and large-hail days (green bars) across Europe: 2000–2020. These quantities are not divided by the number of years because of the incomplete data for the year 2020.**

The percentage of hail days by month per country (for countries with 100 or more reports) for the period
2000–2020 is shown in Fig. 4. Greece is the only country to not have over 50% of its reports being within the
months of May, June, and July, having a more consistent number of hail days throughout the year. Many countries
do not have any reports before April or after September. Spain, Italy, France, and Croatia have similar distributions
of hail days throughout the year, which may be linked to their Mediterranean setting, although Slovenia, Bosnia
and Herzegovina, and Bulgaria do not share the same characteristics, despite also being situated along the
Mediterranean. Previous studies such as Tazarek et al. (2020) have investigated hail distribution in Europe by
linking events to meteorological and climatological factors, which may help explain some of the differences seen
in Fig. 4. Furthermore, Sanchez et al. (2017) investigated hail events in southern Europe, concluding that even
small geographical and climatological differences can have a large impact on the number of hail days reported,
which may also explain some of the differences in Fig. 4.

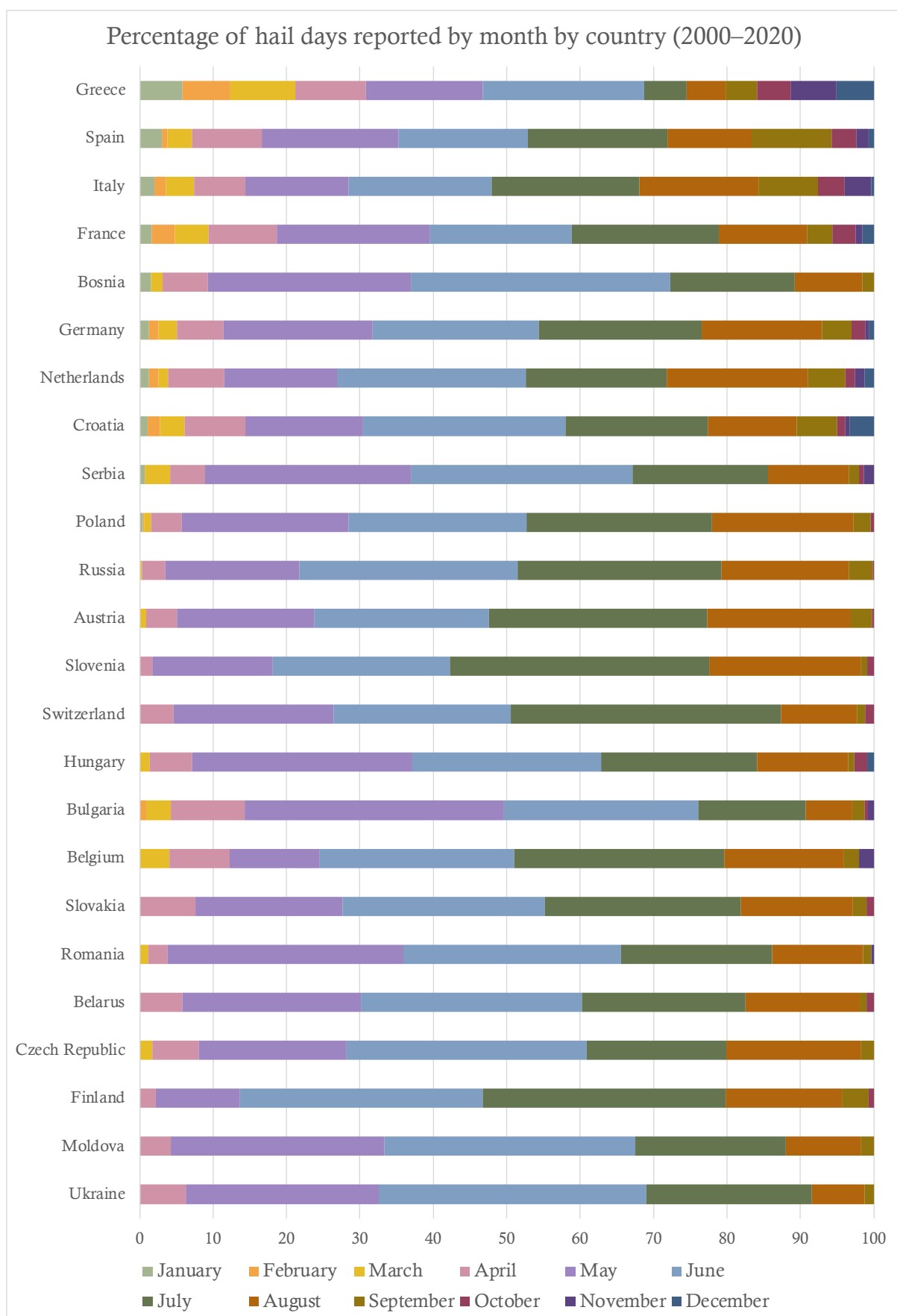

**Figure 4. Horizontal bar charts of the monthly distributions of large-hail reports (%) for countries with 100 or more reports: 2000–2020.**

The average diurnal cycle for the number of large-hail reports between 2000 and 2020 is shown in Fig. 5. The hour 1500–1559 UTC (labelled 1500 UTC) was the most common time for large hail to be reported with a gentle rise and a slightly more rapid decline. When corrected for local legal time (LT) based on each country's official time zone, this peak shifts to 1700–1759 LT because most of Europe is east of the Prime Meridian. Figure 5 can be compared to Púčik et al. (2019, their Fig. 5), who also found a peak during the 1500-UTC hour. These distributions are also similar to those from Kunz et al. (2020, their Fig. 2b) who found a peak during 1500–1800 LT for hailstorms in central Europe using all quality levels from the ESWD, although small differences (e.g., relatively more hail during 1200–1500 LT in Kunz et al. (2020) compared to Fig. 5) may be due to the different study areas between these two studies. Thus, the QC0+ data over a longer period of time used in this study produces a similar climatology and is consistent with previously published research using a shorter period and more selective quality-control levels, indicating that this larger dataset is a reliable source of large-hail data.

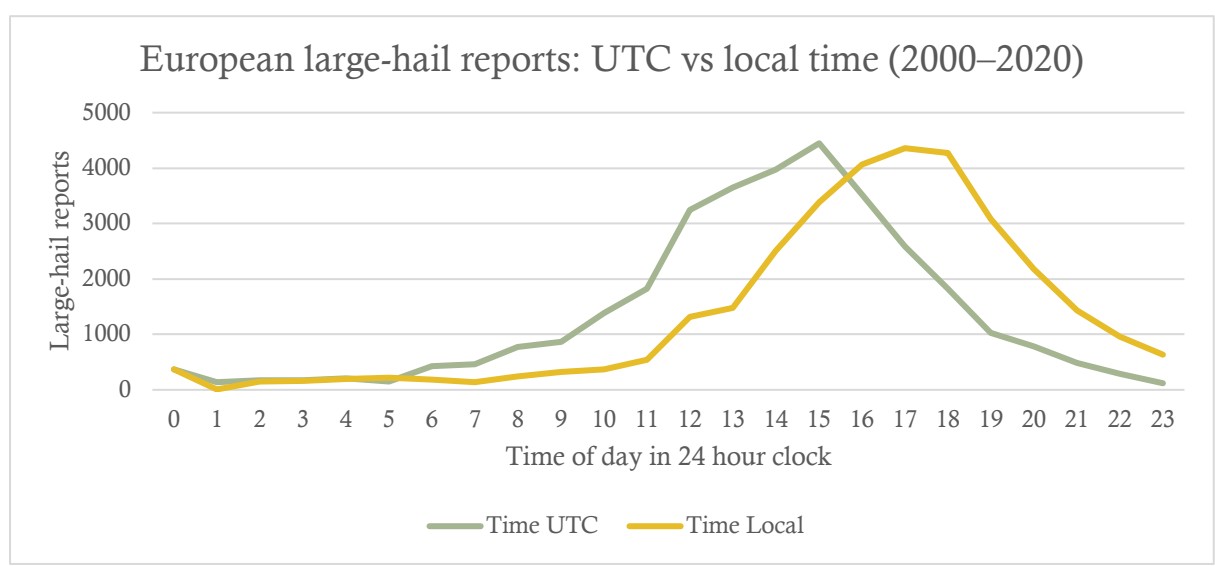

**Figure 5. Distribution of the hourly time of large-hail reports across Europe in UTC (green line) and local time (orange line): 2000–2020. Reports are associated with the starting hour (i.e., a report at 1515 UTC would be placed in the 1500-UTC bin).**

To examine the year-by-year consistency of the diurnal cycle, the distribution of large-hail reports as a
function of local time for each year during the period 2000–2020 is plotted in Fig. 6. Each year mostly reproduces
the diurnal cycle seen in Fig. 5.  The exception is some years, particularly early during this period, that have
unusual peaks at 1000–1200 UTC.  These reports are associated with hail events in the early part of the database
that occurred at an unknown time during the night or day and were placed in 0000 UTC or 1200 UTC, respectively
(Púčik et al. 2019, p. 3906).  However, by 2010, the diurnal distributions seemed to have settled down to look like
that in Fig. 5.  The consistency after 2010 suggests the possibility that the dataset becomes more consistent in
reporting events and could represent a stable period for documenting the present large-hail climate of Europe.


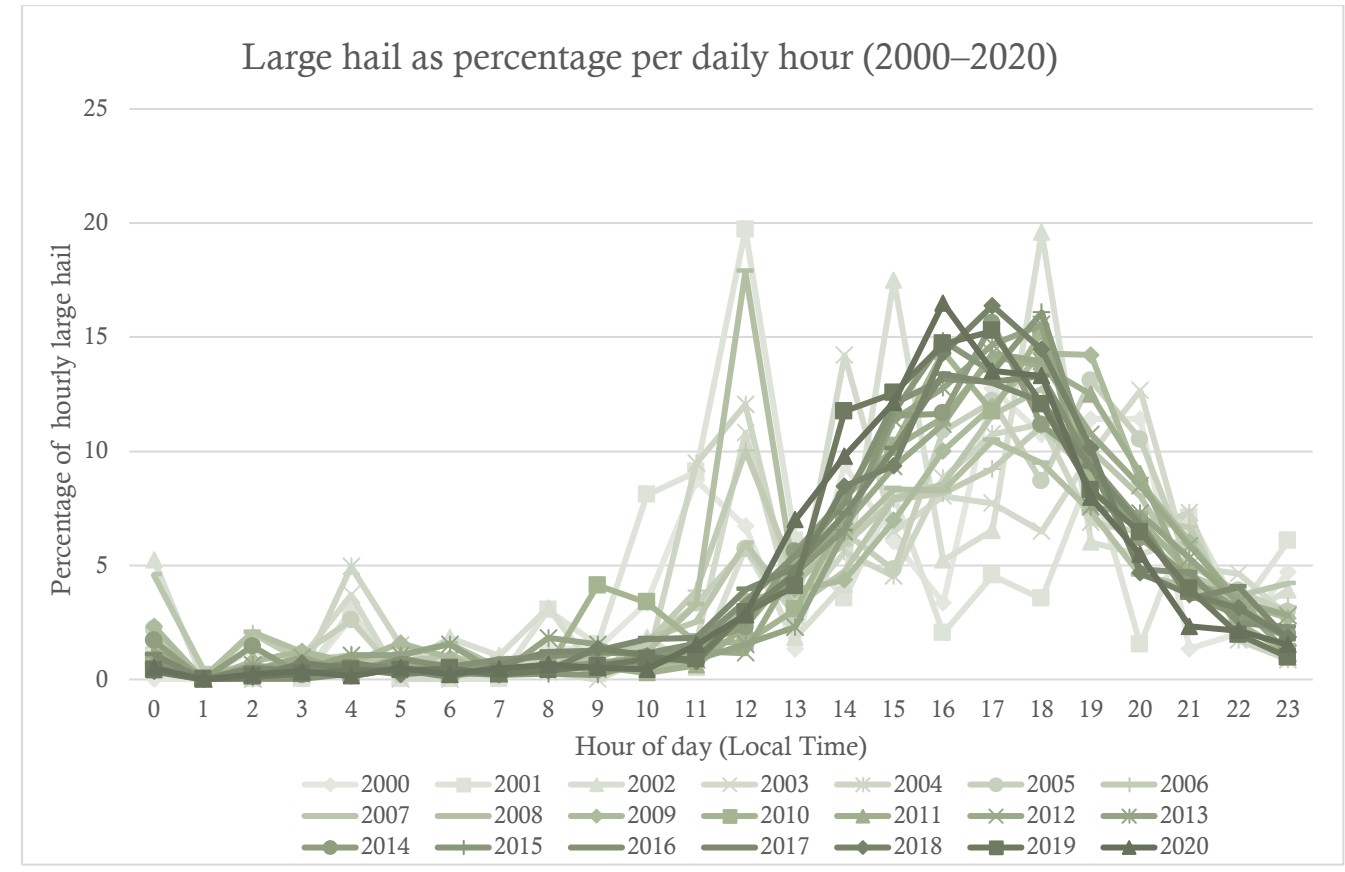


**Figure 6. Hourly percentage of large hail in local time across Europe in local time for each year 2000 to 2020.**


The diurnal distribution by country was also investigated for countries with 100 or more reports (Fig. 7). For
most countries, the time period with the most hail reports is between 1400 and 1800 LT, with little variation
between east and west, and north and south Europe. Belgium seems to be the exception with a larger spread of
times, but has the lowest number of reports out of these countries, with only 121 reports for 49 hail days (Table
1), which is likely not representative of the meteorological conditions that would favor large-hail production.

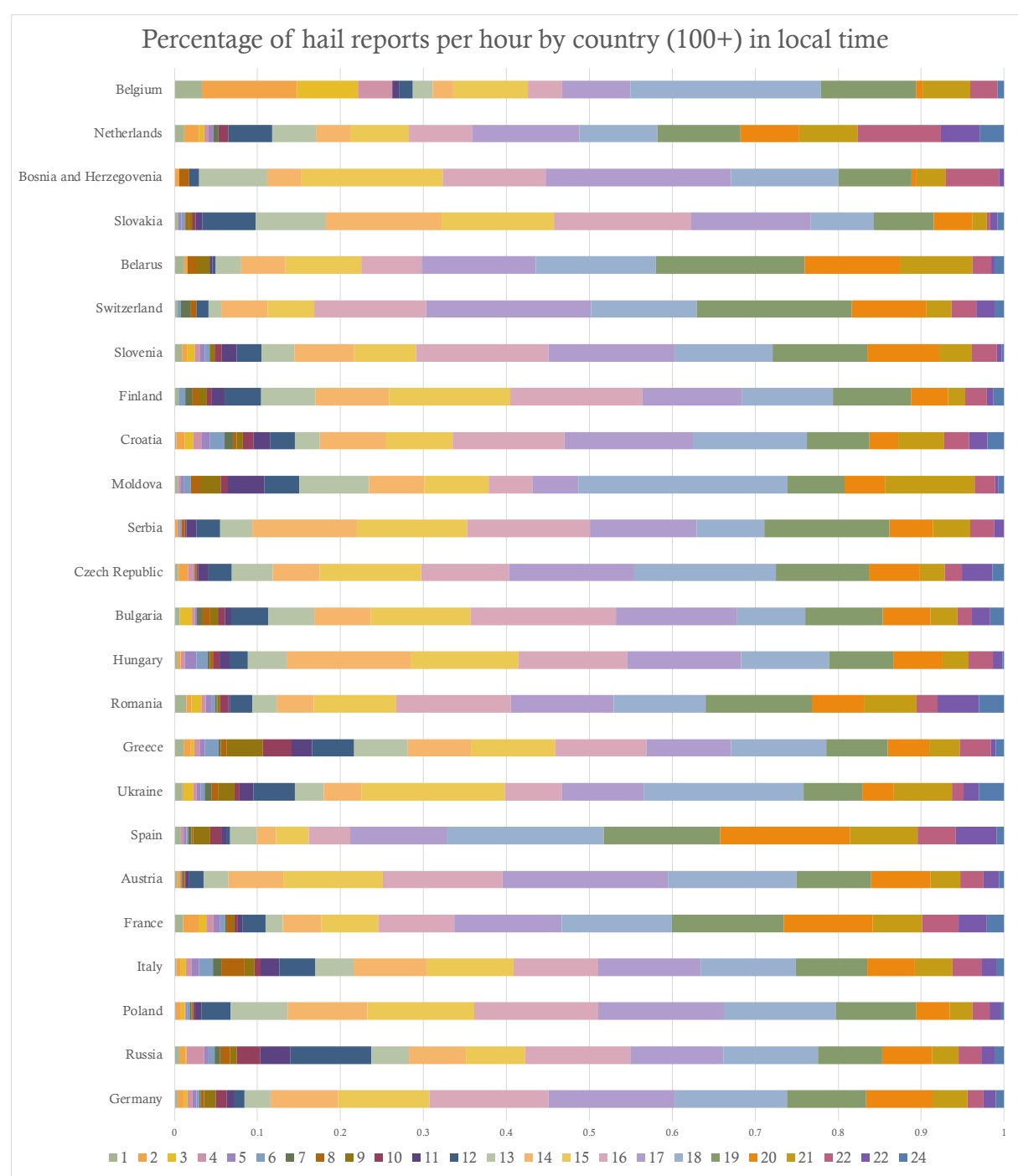


**Figure 7. Horizontal bar charts of the hourly distributions of large-hail reports (percentage divided by 100) for countries with 100 or more reports: 2000–2020.**


**4 Intensity of large hail: 2000–2020**

It is not just the frequency of events that determines their impact on society, but also the intensity of the events, here represented by the maximum diameter of the hail associated with each report. Maximum hail size can be difficult to measure for several reasons as highlighted by Pilorz (2015). For example, as hail is often irregular in shape, the maximum diameter is actually the longest axis of the stone. Therefore, if a stone were more spherical, then its maximum diameter would be smaller than an oblate stone, even though it would have a larger volume. Furthermore, there is always the possibility that the largest hailstone from any given event has not been found or that it has partially melted before discovery.

For the 28,225 large-hail reports in the present study between 2000 and 2020, 18,132 (64%) had data for the maximum diameter. These reports were organized into 1-cm bins, ranging from 2.0–2.9 cm to 10+ cm. Frequency of hail reports decreased with increasing hail size (Fig. 8). The maximum hail size in the database from 2000 to 2020 was 15 cm and was reported in Romania on 26 May 2016. This report was rated QC1, so has been confirmed. The second largest hail size was 14.1 cm and was reported in Germany on 6 August 2013. This particular hailstone set the record for the largest hailstone in Germany (ESKP 2013). This report is recorded as QC2 and includes additional information in the ESWD database, such as the average hailstone size being 8 cm.

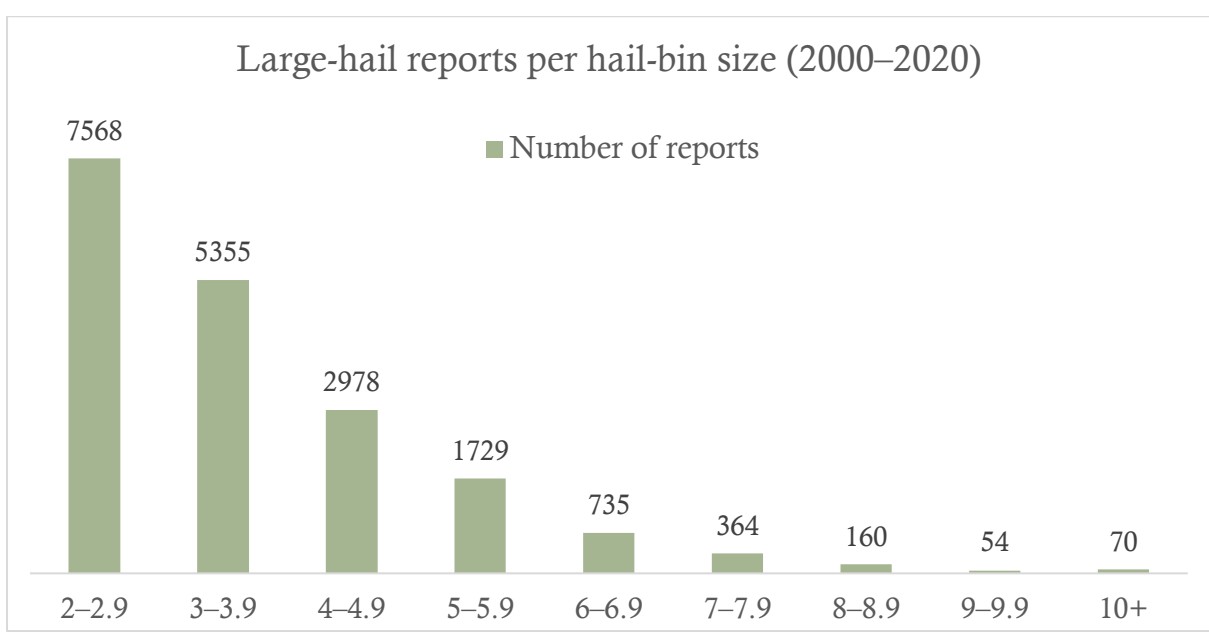

**Figure 8. Bar chart of the number of large-hail reports across Europe by maximum diameter in 1-cm bins: 2000–2020.**

          To investigate the distribution of large-hail size over time, Fig. 9 presents the percentage of each hail-size bin

    per year from 2000 to 2020. During this 21-yr period, the percentage of each bin size does not change dramatically.

    This distribution is similar to the 1989–2018 average from Púčik et al. (2019, their Fig. 7), with about 40% of

    large-hail reports being smaller than 3 cm, about 70% being smaller than 4 cm, and about 84% being smaller than

    5 cm.  Therefore, the large-hail size distribution during 2000–2020 may represent a period of stability in reporting

    with little detectable change in large-hail size distributions in the ESWD dataset. For determining the present

    large-hail climate, the stability in the large-hail size distribution after 2000 represents a slightly longer period of

    record compared to that of the diurnal cycle, which stabilized after 2010 (Fig. 6).

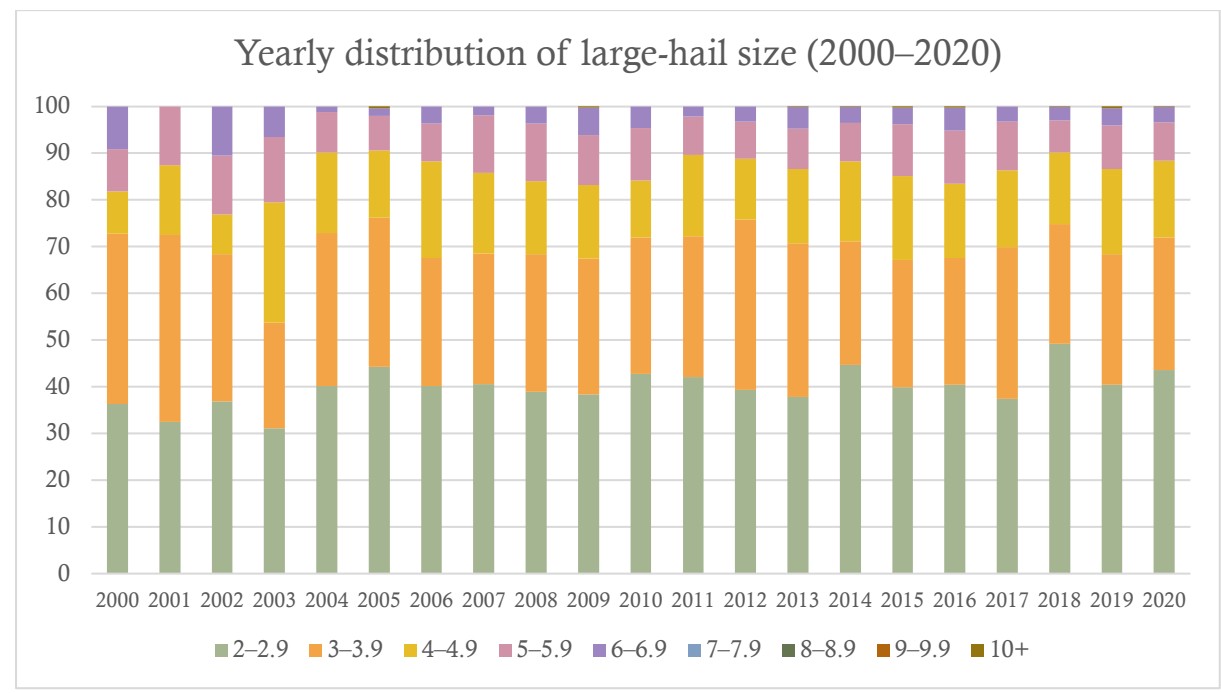

    **Figure 9. Time series of bar charts of the annual distributions of large-hail size across Europe in 1-cm**

    **diameter bins (%): 2000–2020.**

The ESWD has information on average hail size, although only 12% (2237 out of 18,132) of reports contain
this information for 2000–2020. There is, however, a strong positive linear relationship between the average and
maximum hail size recorded (Fig. 10). There were two outliers that are most likely data-entry errors, such as
events with a 2-cm maximum size and 5-cm or 3-cm average size. Both were QC1. The linear relationship ($R^2 =$
0.76) between maximum and average hail size suggests that the average hail size is about 60% of the maximum
hail size, although there is considerable spread around this line.

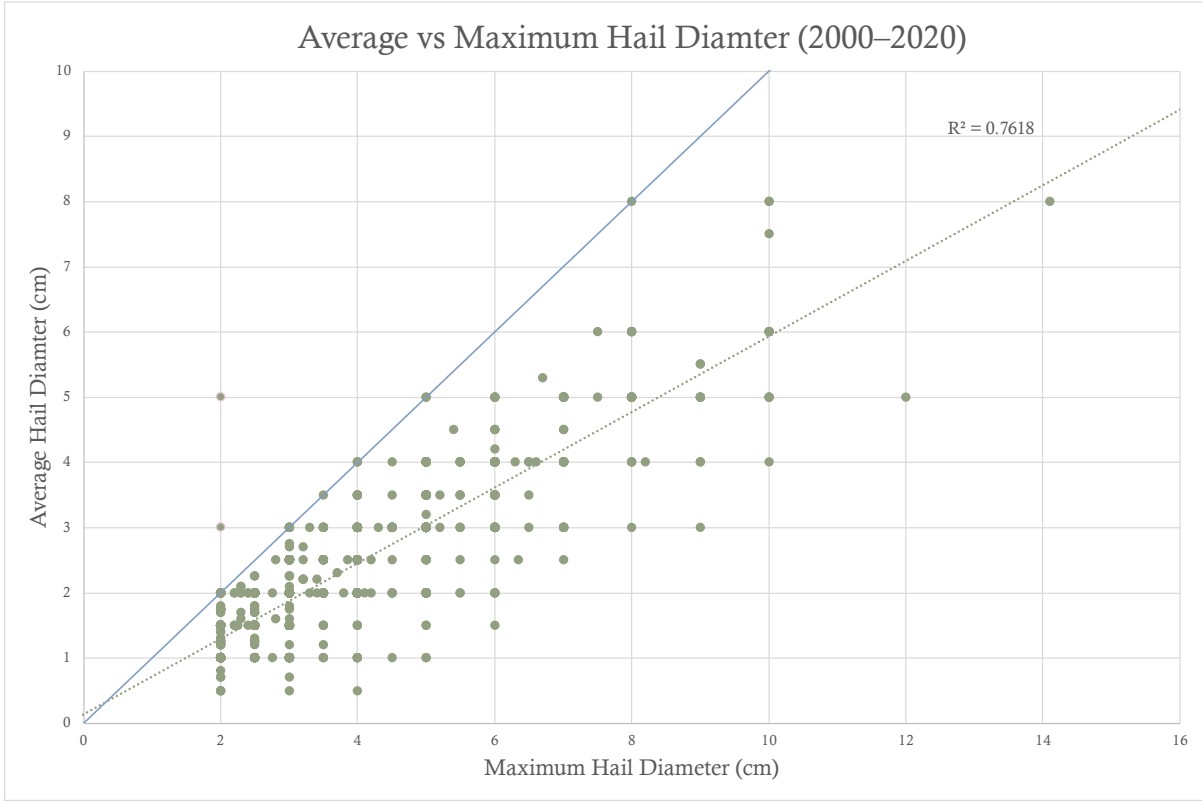

**Figure 10. Scatterplot representing 2237 hail reports of the maximum large-hail size versus average large-**
**hail size across Europe during 2000–2020, with corresponding linear regression line (green dotted line).**
**The 1:1 line is plotted as a blue line. Two pink dots represent likely data-entry errors where the average**
**diameter is greater than the maximum diameter.**

**5 Time accuracy of reports: 2000–2020**

The ESWD includes a quantity called time accuracy, defined as the time interval over which the report could have occurred. For example, a time accuracy of 5 min would mean that the large hail fell within 2.5 min on either side of the time recorded in the ESWD. Groenemeijer and Liang (2020) specify ten categories of time accuracy: 1 min, 5 min, 15 min, 30 min, 1 h, 3 h, 6 h, 12 h, 1 day, and greater than 1 day. The time accuracy of large hail in the ESWD has improved over time, with over 50% of reports having a time accuracy of 30 min by 2012, followed by 50% having a time accuracy of 15 min by 2017 (Fig. 11). Moreover, between 2009 and 2010, reports with a time accuracy of 30 min became more common, replacing some of the reports with time accuracy of 1 h, and time accuracy of 12 h and greater become negligible. Viewing the ESWD from 2000–2020 as a whole, these improvements in time accuracy means that the ESWD is becoming a more reliable source of data, with more highly temporally resolved data on hail occurrence.

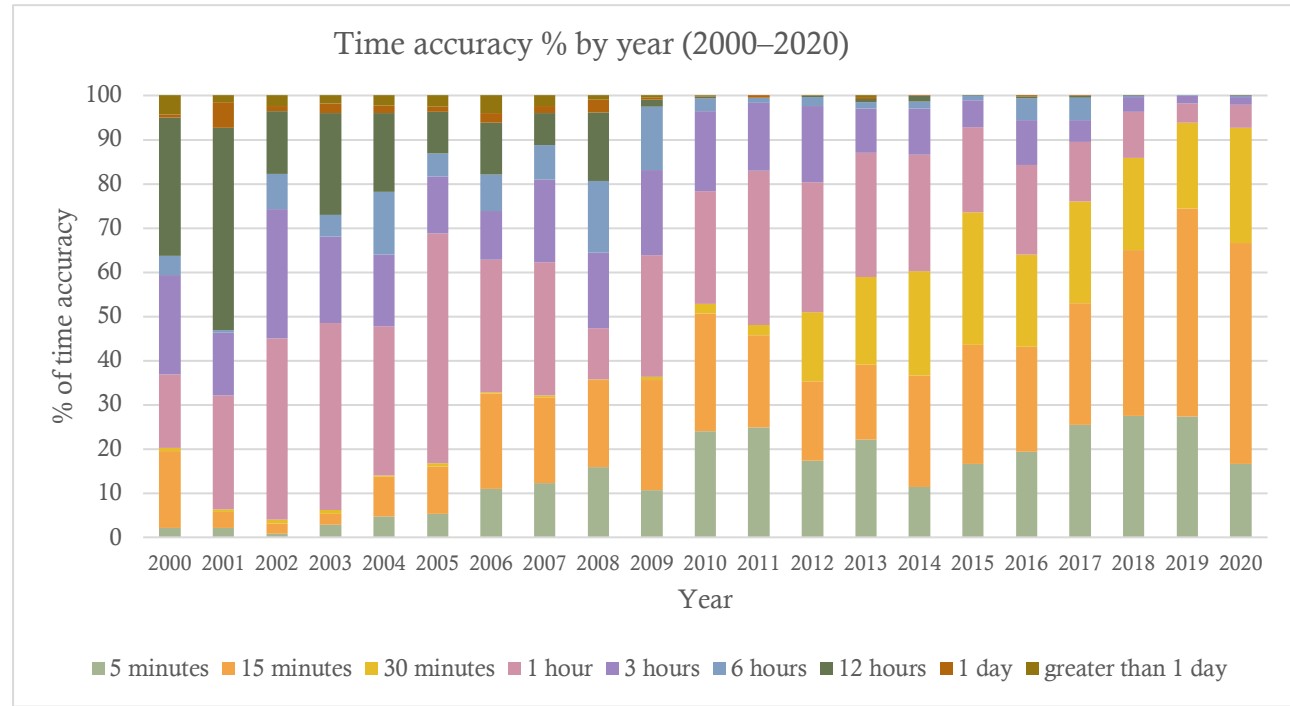

**Figure 11. Time series of bar charts of the annual distributions of time accuracy of reports across Europe (%): 2000–2020.**

On the scale of individual countries, however, work remains to improve the quality of the ESWD. The average time accuracy for each country with 100 or more reports during 2000–2020 is shown in Fig. 12. The distribution of time accuracy varies considerably among these 24 countries. Germany, Finland, and the Czech Republic have more than 40% of their reports with time accuracy of 5 min, whereas Bulgaria, Russian Federation, and Moldova have the lowest (1% or less). Figure 12 also indicates the countries for which there is opportunity to improve engagement in severe-weather reporting.

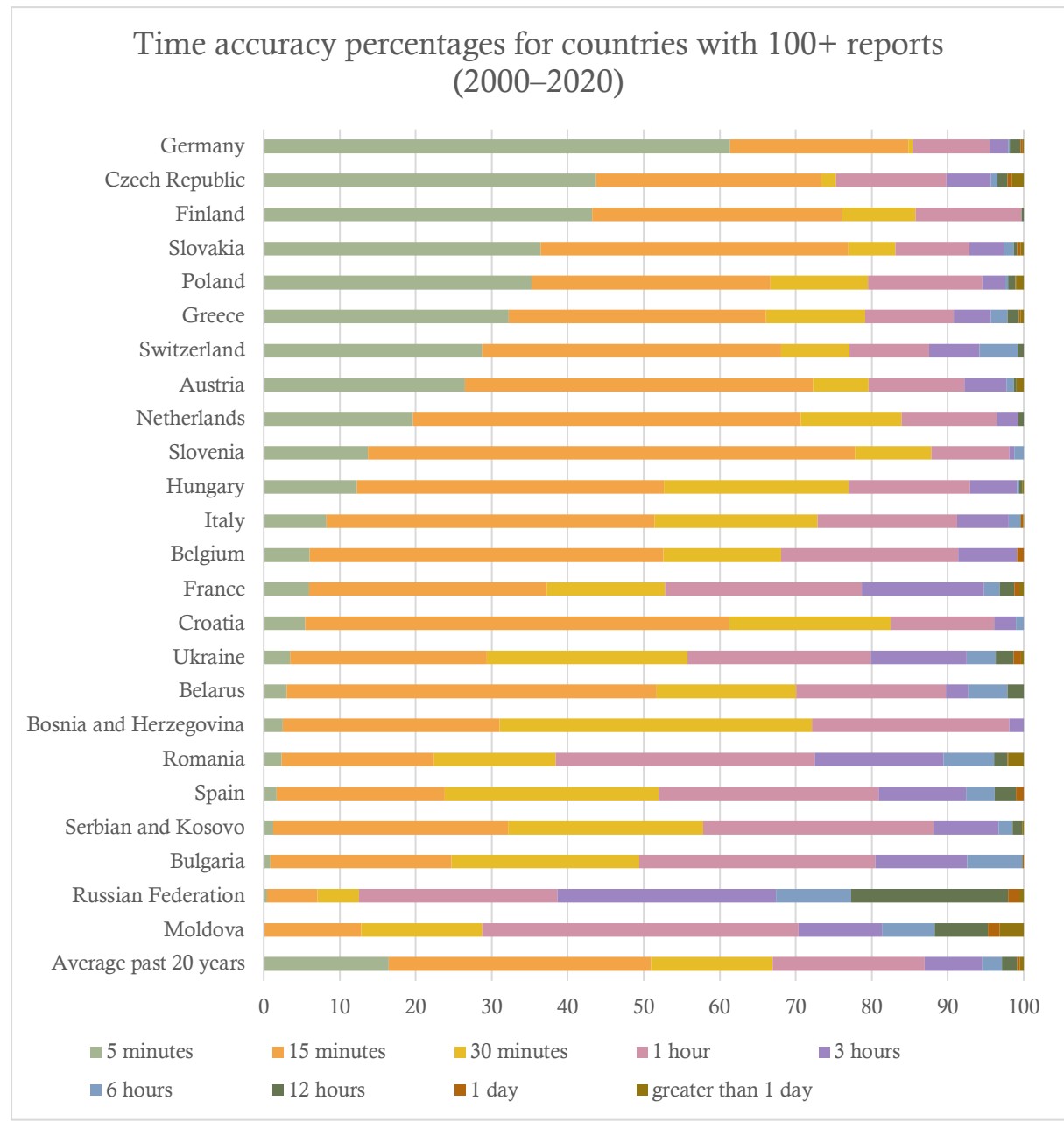

**Figure 12. Horizontal bar charts of the time accuracy for countries with 100 or more reports (%): 2000–2020.**

## 6 Spatial distribution by country: 2000–2020

Hail reports across Europe are heterogenous, not just in time, but also in space. Countries such as Germany, Russian Federation, and Italy reported 4956, 4182, and 2447 large-hail events between 2000 and 2020, compared to others such as Switzerland, the UK, and Denmark only reporting 266, 85 and 31 cases, respectively (Table 1). Central and western European countries reported more large hail with 5 out of the top 10 countries located there (Table 1). Germany has more large-hail reports than the Russian Federation for fewer large-hail days, similarly to Poland having more reports than Italy, and Austria more reports than Greece. The ESWD grew out of other data-collecting efforts such as TorDACH (i.e., a tornado dataset collection effort from Germany, Austria, and Switzerland), which may partially explain why there are more reports for a similar amount of days in Germany, and Poland has a long history of hail reports (section 7).

Besides meteorological reasons for the variability, other reasons that may explain these reporting differences include the existence, size, and enthusiasm of spotter networks within each country; variations in the ability or enthusiasm of citizens to input into the ESWD; and the availability of information to quality-control reports. In fact, many central European countries have larger and more enthusiastic spotter networks [e.g., Poland, as discussed in Pacey et al. (2021) and section 7 of the present article] and are more likely to enter their reports into the ESWD. KERAUNOS, based in France, or the MeteoSwiss app based in Switzerland, for example, also encourage citizen involvement in reporting of extreme events, which are imputed into the ESWD database. Population density and area of the country were considered as possible explanations for the number of hail reports varying by country, although neither had a statistically significant relationship with the number of hail reports (not shown). As with the time-accuracy data (section 5), greater engagement with some countries to encourage entering their reports into the ESWD would lead to a larger and more complete dataset.

348 **Table 1. Number of large-hail days and large-hail reports by country: 2000–2020.**

| Country | Number of large-hail reports | Number of large-hail days |
|---|---|---|
| Germany | 4956 | 692 |
| Russian Federation | 4182 | 1012 |
| Poland | 3226 | 471 |
| Italy | 2447 | 555 |
| France | 1707 | 440 |
| Austria | 1502 | 353 |
| Spain | 1027 | 295 |
| Ukraine | 1021 | 319 |
| Romania | 983 | 267 |
| Greece | 975 | 395 |
| Hungary | 903 | 226 |
| Bulgaria | 820 | 238 |
| Serbia and Kosovo | 490 | 146 |
| Czech Republic | 490 | 174 |
| Moldova | 451 | 117 |
| Croatia | 399 | 181 |
| Finland | 382 | 139 |
| Slovenia | 332 | 116 |
| Switzerland | 266 | 87 |
| Belarus | 261 | 103 |
| Slovakia | 234 | 104 |
| Bosnia and Herzegovina | 169 | 65 |
| Netherlands | 165 | 76 |
| Belgium | 121 | 49 |
| Latvia | 86 | 50 |
| United Kingdom | 85 | 41 |
| Estonia | 79 | 38 |
| Portugal | 77 | 34 |
| Sweden | 74 | 50 |
| Cyprus | 68 | 45 |
| Lithuania | 42 | 23 |
| Luxembourg | 39 | 6 |
| Denmark | 31 | 18 |
| Albania | 22 | 12 |
| Montenegro | 21 | 3 |
| North Macedonia | 21 | 13 |
| Norway | 21 | 15 |
| Malta | 11 | 9 |
| Andorra | 6 | 4 |
| Iceland | 4 | 4 |
| Ireland | 2 | 2 |

349

Similar to Fig. 2 where the number of large-hail reports was plotted versus the number of large-hail days by year, Fig. 13 shows a scatterplot between the number of large-hail reports versus the number of large-hail days by country from Table 1. There is a positive linear relationship ($R^2 = 0.88$) between large-hail reports and large-hail days by country (Fig. 13), suggesting that large-hail reports are proportional to large-hail days. This relationship would therefore imply that reporting frequency is similar across all hail frequencies and countries, except for Germany and Poland which have a much greater number of reports proportional to the number of days.

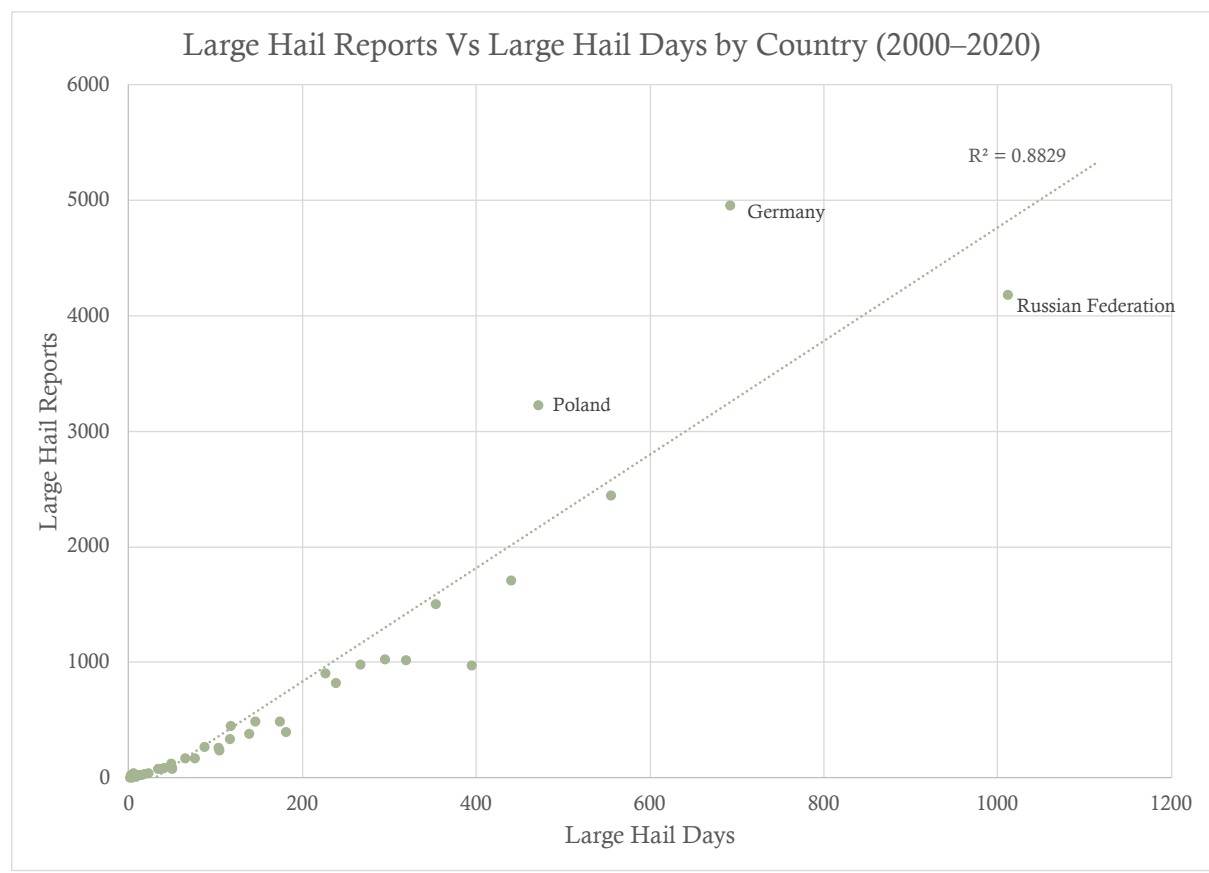

**Figure 13. Scatterplot of the total number of large-hail reports versus large-hail days by country: 2000–2020.**

We also investigated the number of large-hail days for each country with 100+ reports for the period 2000—2020 (Fig 14 a, b, c, and d). We separated these countries into 4 groups based on their total number of large-hail days for ease of visualization. We do note that 2020 may show slightly fewer large-hail days than other years since the last 3 months of the year are omitted from this dataset.

Figure 14 (a) shows the number of large-hail days per country for the top 6 countries with 100+ reports for the period 2000–2020. In this subset, Greece displays the fewest large-hail days with 395 days, and Russia the greatest, with 1012 days. Germany appears to have the most stable number of annual large-hail days over this period, notably from 2003 onwards. However, there remains some year-to-year variation, with 2003–2009 showing the most stable period. Russia also shows a consistently high number of annual large-hail days throughout this period, and although there is a lot of variation up until 2013, the number of large-hail days appears to stabilise after this. Italy shows a steady increase in large-hail days up until 2014, after which a slight decline is seen before rising again from 2016 onwards. France, Poland, and Greece all appear to see a rise in large-hail days from around

2005 onwards, with Poland showing a consistent number of large-hail days from then on, while large variability is still seen in France and Greece.

Figure 14 (b) shows the number of large-hail days per country for the upper middle 6 countries with 100+ reports for the period 2000–2020. In this subset, Hungary displays the fewest large-hail days with 226 days, and Austria the greatest, with 353 days. Out of the 4 groups, this one shows the most consistent and significant rise in large-hail days over this period, although there remains much annual variation for each country. Ukraine displays an anomalous spike of 40 large-hail days in 2002, a total which is not again reached over this period, with the second highest large-hail year being 2019 with 36 days. Bulgaria and Hungary have a similar number of large-hail days throughout this period, these gradually increasing until 2016, after which they start to decline. Additionally, with the exception of 2012, Austria shows a consistent number of large-hail days for the 2010–2018 period.

Figure 14 (c) shows the number of large-hail days per country for the lower middle 6 countries with 100+ reports for the period 2000–2020. In this subset, Slovenia displays the fewest hail days with 116 days, and Croatia the greatest, with 181 days. This group shows significant year-to-year variation in large-hail days, notably for Slovenia, Moldova, and Finland. Finland has the largest variation in annual large-hail days in this group, with 22 reports in 2008, and none in 2012 and 2015. Serbia and Kosovo, The Czech Republic, and Croatia have a similar number of large-hail days over this period, although between 2006–2008 and then again from 2019, they display a greater difference. Slovenia has seen several peaks in large-hail days, the first being in 2005, followed by the greatest peak in 2009 with 18 large-hail days, which was then followed by a quick decline, before increasing again from 2015 onwards. Moldova initially demonstrated a steady increase in large-hail days, followed by a peak between 2013 and 2016, with 2014 seeing the greatest number of large-hail days here with 24 days.

Figure 14 (d) shows the number of large-hail days per country for the bottom 6 countries with 100+ reports for the period 2000–2020. In this subset, Belgium displays the fewest hail days with 49 days, and Slovakia the greatest, with 104 days. There appears to be a rising trend in the number of large-hail days reported for each country. However, as these countries have few annual large-hail days, it is difficult to determine whether this rise is due to increased reporting or an increase in large-hail events. Furthermore, although all countries exhibit annual variation, Belarus shows the greatest variation in this group.

Although there generally is a rise in the number of large-hail days for each country throughout the period 2000–2020, there remains much annual variation. The top 50% of countries with 100+ reports for this period are mostly showing more consistency in the number of annual large-hail days than the bottom 50%. However, the bottom 25% of countries are generally showing an increase in annual hail-days for this period, although it is difficult to assess any real trends in large-hail days as these may be due to a better reporting, and not more large-hail events.

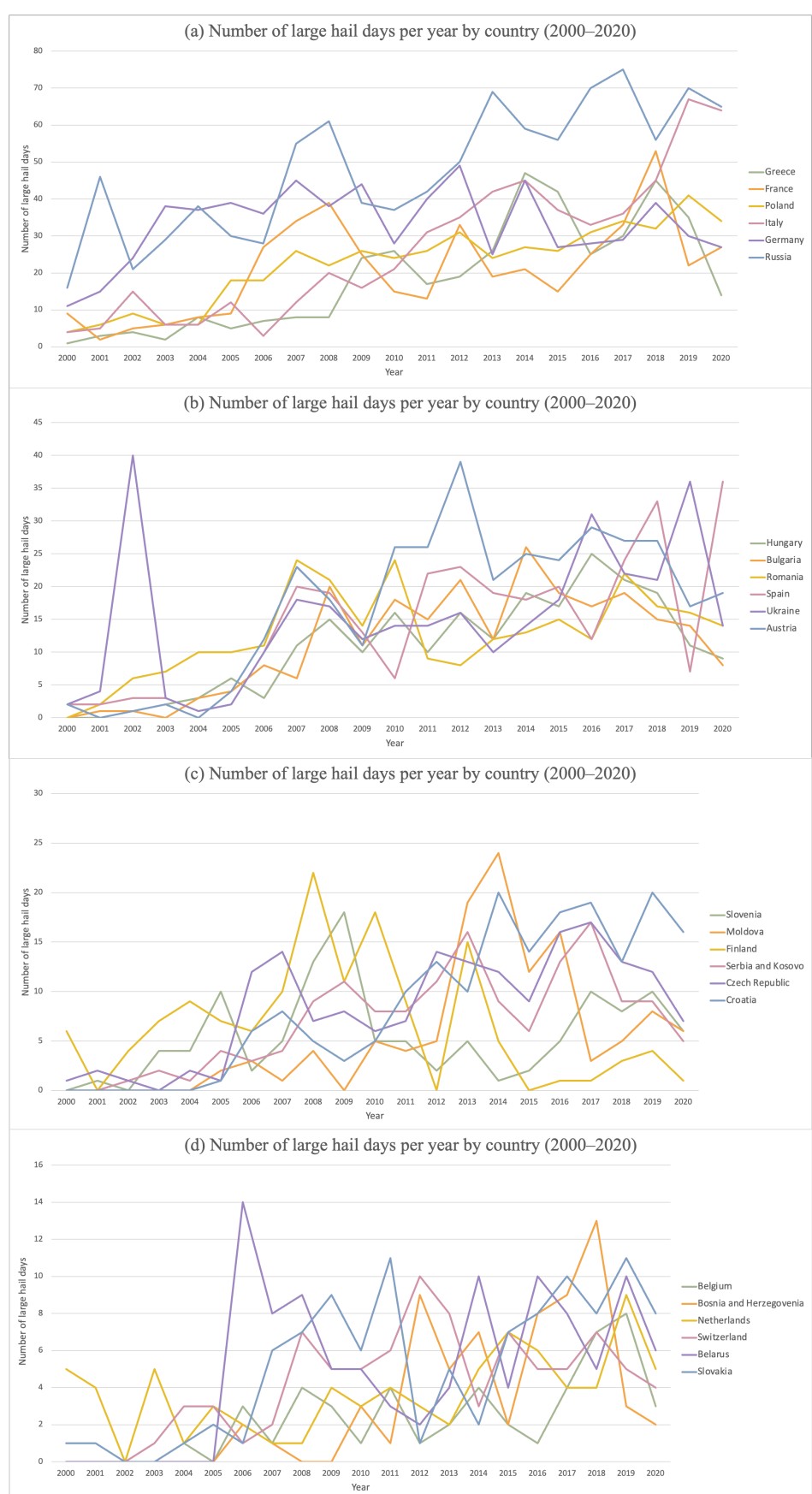

405

**Figure 14. Line graph of large hail days per country for countries with 100+ reports: 2000–2020. (a) top 6**

**countries, (b) upper middle 6 countries, (c) lower middle 6 countries, and (d) bottom 6 countries.**

406

407

We further investigated the hail-size distributions by country for the period 2000–2020 (Fig. 15). Only one report of each size diameter was taken per country per day to minimize some of the reporting biases. Finland has the greatest proportion of the lowest hail bin size, whereas Slovenia has the lowest. For sizes 5 cm in diameter and greater, the proportion of hail sizes recorded starts to diminish drastically, which would be expected as larger hailstones are rarer. Although Slovenia has the greatest proportion of hail sizes above 5 cm, these reports came from a sample of 116 hail reports, one of the smallest of the countries analyzed. For hail days with a report above 10 cm, Russia has the greatest quantity with 10 reports over this period, whereas Italy came second with 9 reports and France with 8. Slovenia, although having a greater proportion, had 5 days with a hail report above 10 cm for this period.

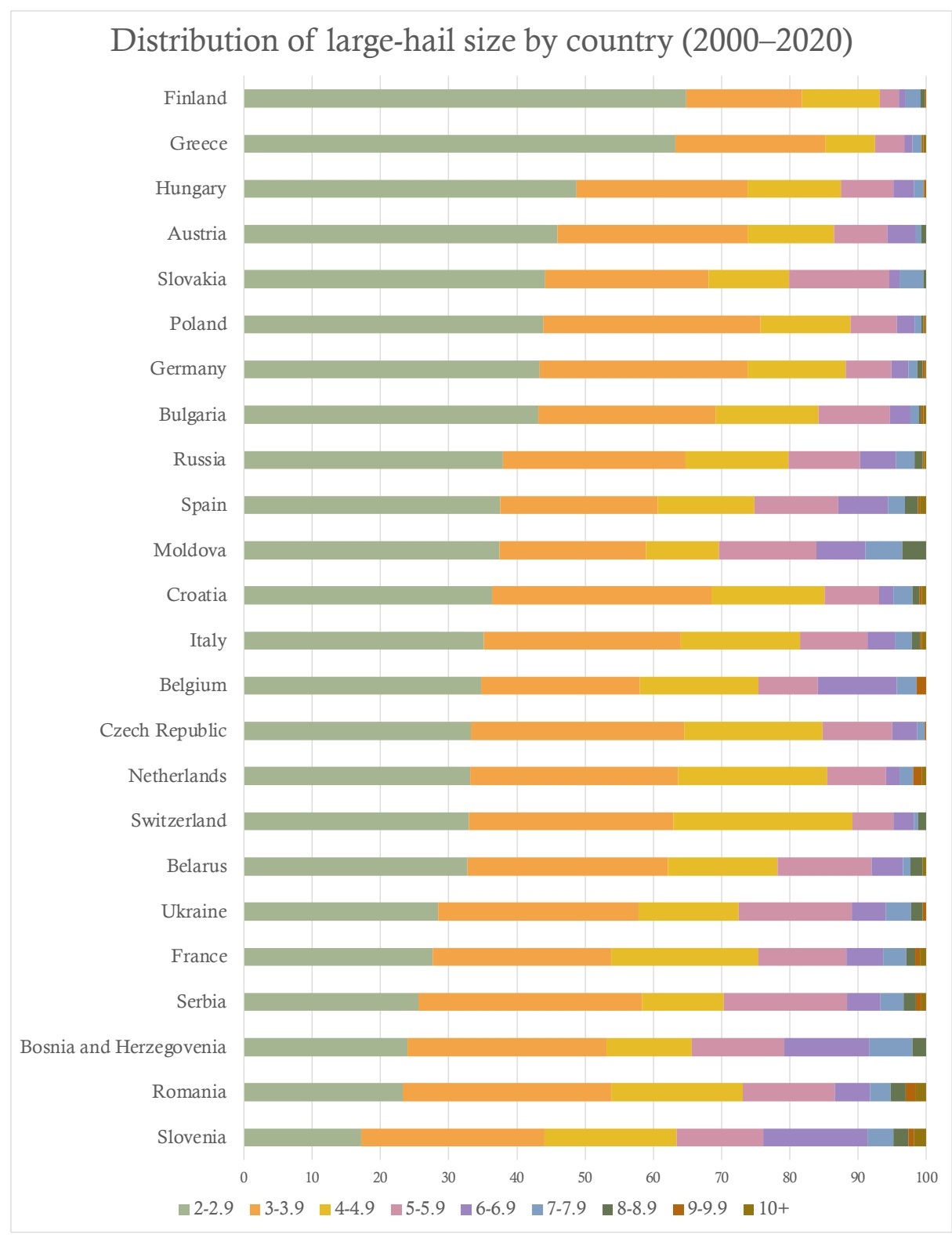

**Figure 15. Horizontal bar charts of the size distribution of large hail for countries with 100 or more reports (%): 2000–2020.**

**7 Poland: 1900–2020**

As noted in association with Fig. 1, nearly all large-hail reports and large-hail days during the 1930s and 1940s–1950s originated in Poland (Figs. 16a,b). Very few hail days were recorded between 1956 and 2000, before the general increase along with the rest of Europe for the last 20 years (Fig. 15). There appears to be far fewer large-hail days over the past 20 years in Poland (30–40 days a year) compared to the 1940s–1950s (100–120 days a year). With an overall increase in reporting numbers and accuracy, it would be unlikely that the current Polish reports are missing many events, and therefore the difference in annual numbers of large-hail days seems unlikely.

The addition of this data in the ESWD was due to Igor Laskowski who reports:

> "those reports were based on annual records collected by a Polish National Institute of Meteorology founded in 1919, now Institute of Meteorology and Hydrology - National Research Institute (https://imgw.pl/instytut/historia). The data was collected via hail questionnaires, which provided information on the size of the hail (vetch-sized, pea-sized, broad bean-sized, hazelnut-sized, walnut-sized, pigeon egg-sized, hen egg-sized and goose egg-sized) and also details about time of its occurrence, storm direction and the size of the expected yield decrease (in percent). The questionnaires were filled in both by agricultural correspondents of the Polish Central Statistical Office (whose number was growing larger, especially in the [19]50s) and existing insurance companies which provided hail insurance at this time. Those records also contain observations of hail reported by observers at meteorological stations."

At the time of this study, data from yearbooks from 1930–1937 and 1946–1955 had been added.

Suwała (2011) investigated Polish hail based on data from 23 meteorological stations recorded in the Meteorological Yearbooks published by the Institute of Meteorology and Water Management for the years 1973–1980 and the Polish National Climatic Data Centre for the years 1981–2009. They found that over the 37-year period, March was the month with the highest hail frequency across the country, followed by February and January. For individual stations, December and January recorded the highest hailfall, with the two stations along the Baltic coasts having a mean of 8 days. Although these results may indicate a cool-season preference for hail, there is the possibility that ice pellets or graupel might have been classified as hail (e.g., Punge and Kunz 2018). Overall, the Baltic coast showed the highest annual mean, whereas central Poland showed the lowest. This result contradicts the findings of Pilorz (2015) who investigated large hail in Poland for 2007–2015, concluding that southeast Poland had the greatest number of storms and associated large hail events.

Furthermore, the warm months of June to September had the lowest mean hail frequency for all stations. This contradicts the results found in this present study and those by Púčik et al. (2019) that hail is most frequent in the warm season, but also contradicts those by Taszarek and Suwała (2014) who investigated large hail in Poland in 2012. In addition, there appeared to be some cyclicity in frequency over the 37-yr period, although this cyclicity varied greatly when investigating individual stations, and no trends were observed. These results may explain why Poland possesses a different annual distribution to other locations.

Suwała (2011) mentioned previous hail studies in Poland, such Schmuck (1949), Koźmiński (1964), and Zinkiewicz and Michna (1955), which may offer an explanation on the high number of hail reports during the 1930s and 1950s. Unfortunately, these are not currently available to read. Access to these historical studies may help explain the quantity of Polish entries in the ESWD during the 1930s, 1940s, and 1950s. Moreover, an effort to retrieve and input the data from 1973 to 2009 into the ESWD would greatly help with the homogeneity of the

Polish dataset. There remains the possibility that this data does not exist as the country suffered major economic
difficulties during this period.

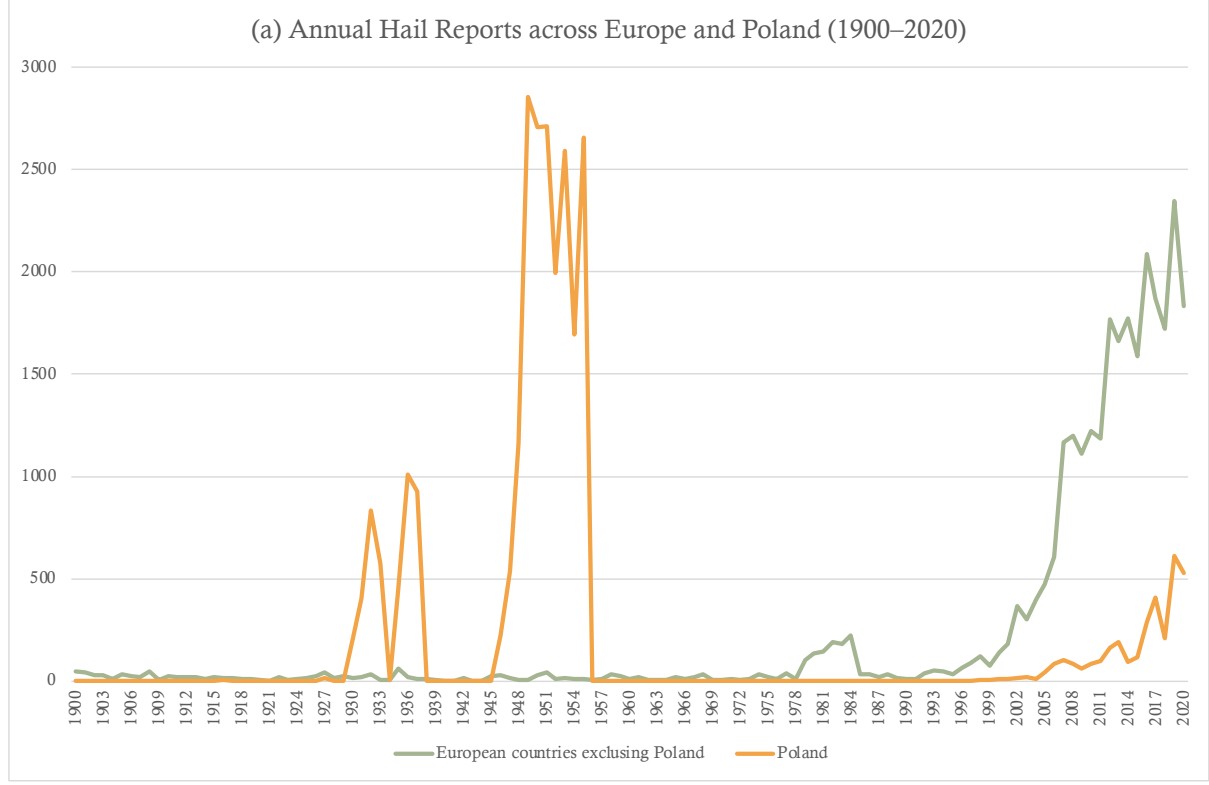


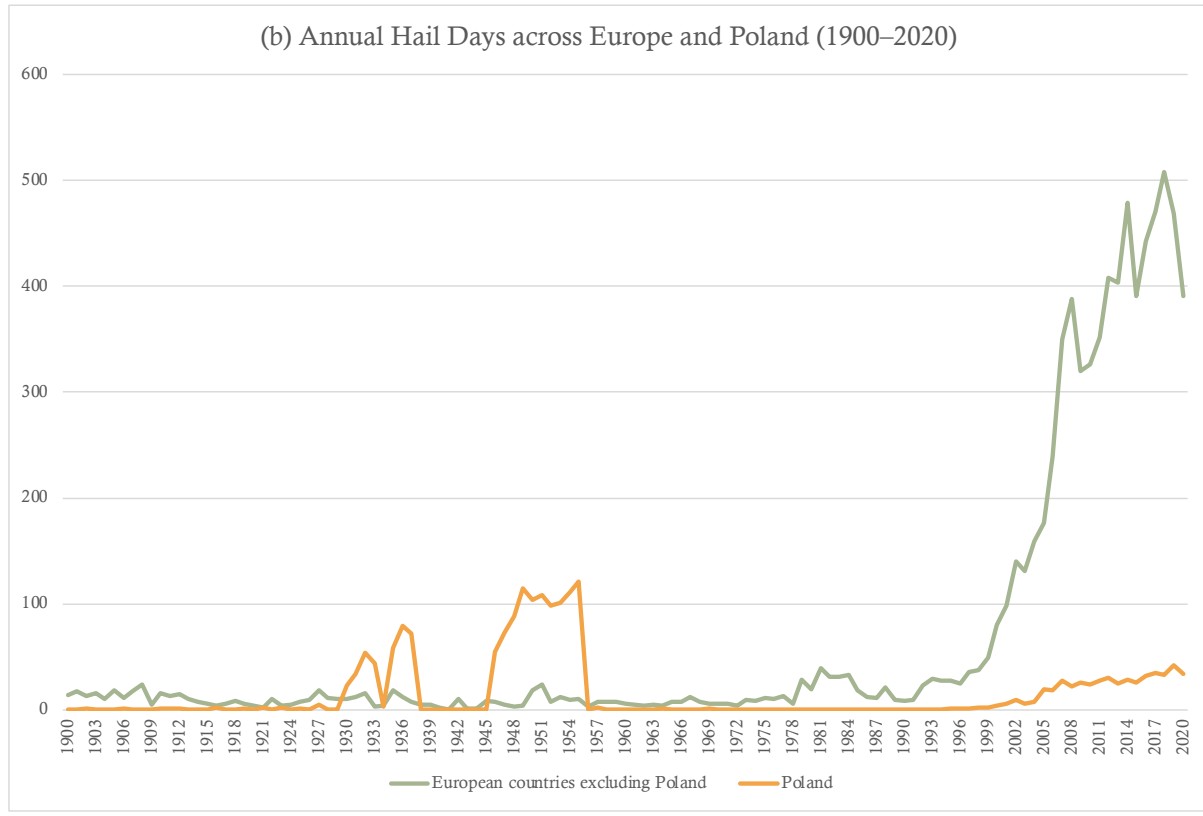


**Figure 16. Time series of annual numbers of (a) large-hail reports for Europe (green line) and Poland (red**

**line), and (b) large-hail days for Europe (green line) and Poland (red line): 1900–2020.**

As in Fig. 11, the time accuracy of large-hail reports can be plotted as a function of time during 1930–2020
in Fig. 17.  The time accuracy of reporting in Poland has improved over the past 20 years, with over half the
reports having a time accuracy of 15 min by 2015 (Fig. 16). During the 1930s and 1950s, the time accuracy was
much lower, around 3 h (Fig. 16). Although this result may suggest that reports were less reliable during this
period, the consistency in time accuracy (especially during the 1950s) may also suggest that the data-collection
methods were more consistent. These reports were later found to be based upon the Meteorological Yearbooks
from the Polish National Institute of Meteorology (I. Laskowski 2022, personal communication). The yearbooks
contained information on hail size, time of occurrence and storm direction based upon questionnaires posed to
insurance companies, agricultural correspondents of the Polish Central Statistical Office alongside observations
from meteorological stations. Laskowski also mentioned that yearbooks from the 1960s and 1970s also existed,
but was currently unable to find any existing copies. Hence, such data – when it is found – remains to be entered
into the ESWD.
In addition, the reported location accuracy was also investigated, with the most common distances being 1
and 3 km, similar to those found in the broader 2000–2020 dataset. This result reiterates the importance of these
earlier reports in constructing a reliable hail climatology, and gives credit to the data-collection method.
The historical Polish datasets offer an insight into past hail frequency and reporting accuracies. Results by
Suwała (2011) for the period 1973–2009 contradict those found for more recent time periods in terms of peak
annual frequency and spatial distribution of large hail. The potential implications of these discrepancies may
suggest that distributions of hail size, frequency, and location have changed over time and have not yet been
established or studied due to the lack of historical pan-European data, highlighting the importance of building the
ESWD further. Moreover, the existence of Meteorological Yearbooks in Poland could also suggest that other
nations might hold similar records that remain to be analyzed and could contribute toward building a more
complete climatology.

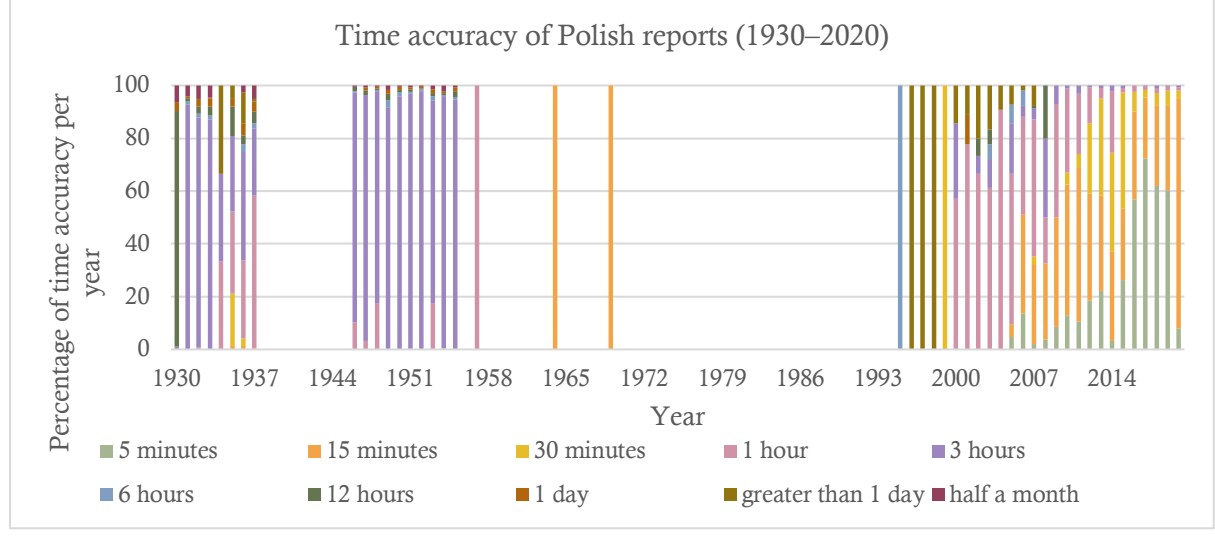


**Figure 17. Time series of bar charts the annual distributions of time accuracy of reports for Poland (%):**
**1930–2020.**

**8 Comparison to previous hail climatologies and prospects for a baseline for climate-change research**

The ultimate goal of severe-storm climatologies is to create a consistent and complete database in space and time. Consistent data acquisition methods throughout the study area and through time would assist in achieving this goal; however, consistency is not achievable across Europe. Punge and Kunz (2016) synthesized all European hail studies in their review, not just large hail. They concluded that not all regions have the same threat of hail, and they found that efforts to report and record these events vary by country. They further concluded that there was insufficient evidence to determine any trends in hail events, both in terms of spatial and temporal extent, highlighting the need for the continuation of the ESWD to form a reliable climatology. Previous studies have provided pan-European climatologies of hail based using other methods such as Punge et al. (2014, 2017) who used overshooting cloud tops, Rädler et al. (2018) who used reanalysis data, or Taszarek et al. (2018) who used a combination of data sources. Some studies projected increases in hailstorms with climate change in Italy (Piani et al. 2005), Netherlands (Botzen et al. 2010), and Germany (Mohr et al. 2015), as well as across much of Europe (Taszarek et al. 2021). Other studies have also concluded that there were no positive trends in the frequency of hail in hailpad data in northern Italy and France (e.g., Eccel et al. 2012; Dessens et al. 2015; Raupach et al. 2021; Manzato et al. 2023). Tazarek et al. (2019) argued that a combination of datasets is important to construct a robust climatology, particularly as the spatial and temporal resolutions would often differ between methods. Furthermore, studies such as Rädler et al. (2018) compared their reanalysis results to surface observed reports from the ESWD to strengthen their arguments. Therefore, understanding the characteristics of the current surface observations via the ESWD helps not only build a climatology of large hail in Europe, but can also be used in association with other research methods to identify the underlying factors which lead to such events.

Examining the evidence presented in the present article, we seek a stable time period during 2000–2020. Based on the number of large-hail reports, no stable time period exists (Fig. 1). Based on the number of large-hail days, the time period starts around 2012 (Fig. 1). Based on the diurnal cycle of large-hail reports, the time period starts around 2010 (Fig. 6). Based on the large-hail size distributions, the time period starts around 2004 (Fig. 9). Based on the time accuracy of reports, the time period possibly starts around 2018 (Fig. 11). However, if one is prepared to accept an accuracy of 3 h or less, then the time period starts around 2010 (Fig. 11).

**9 Conclusion**

The ESWD provides the only pan-European dataset for large-hail reports. The frequency of reports is sporadic pre-2000, and hence the focus of this study is for the period 2000 to 2020. Hail reports have continuously increased since 2000. The annual number of large-hail days have remained steady after 2010 at around 175 days per year, although some interannual variability is still observed. Increased large-hail reports for similar large-hail days suggests that a greater spotter network is in operation, and that the engagement with the ESWD is increasing. When considering the annual number of large-hail days per country, there does appear to be an overall increase in the quantity observed for the countries which previously reported fewer hail-days, while those which observed greater numbers throughout this period seem to be stabilizing.

The warm season of May to August shows the highest number of large-hail reports and large-hail days, with June showing the highest large-hail reports and July the highest large-hail days. The number of large-hail reports decrease faster than large-hail days from June to September. The diurnal cycle shows that the peak hailfall time is 1500 UTC and 1700 local time.

The number of large-hail reports decreases with increasing diameter, and the percentage distribution of each
large-hail size by year does not appear to have changed over the past 20 years. The possibility that hail-size
distribution is changing remains, as smaller, less damaging hail size events are being recorded more regularly.
The diurnal cycle by year shows that for the past 10 years, a consistent pattern has emerged, with a rise in the
early afternoon and a decline in the evening. Furthermore, the time accuracy of reports has improved with over
50% of reports being reported to within a 30-minute window by 2012, followed by 50% being reported to within
a 15-minute window by 2017. Not all countries display improved time accuracies. Germany, Finland, and the
Czech Republic have the greatest proportions of 5-minute time-accuracy reports, whereas Russia, Moldova, and
Bulgaria have the highest proportions of 1-h or greater time-accuracy reports. Efforts to improve monitoring and
reporting in these regions is therefore suggested to improve the completeness of the ESWD.
Poland possessed anomalously large numbers of large-hail reports during the 1930s, 1940s, and 1950s. The
reason is linked to scientific interest in severe convective storms during these periods alongside a nationwide
effort by the Polish National Institute of Meteorology to record hail events via questionnaires. Yearbooks also
exist for the 1960s and 1970s; however, copies are yet to be retrieved and entered into the database.
Even though the dataset remains too short to extract any trends in large-hail pattern distribution, the
climatology presented here provides insight into which countries and geographical regions to target for
improvements in data acquisition. This climatology also helps advance the idea that some time series are starting
to show consistent behavior, suggesting their utility as climate-change baselines. Furthermore, the differences in
both spatial and annual frequencies of hail in Poland over different time periods may suggest that hail trends have
been changing, highlighting the importance of building and maintaining such climatologies. Therefore, the
usefulness of the ESWD will only continue to expand and offer avenues for future severe convective storm
research.

*Data availability.* The data were obtained from the European Severe Storms Laboratory European Severe Weather
Database, in accordance with their data policies: http://www.eswd.eu.

*Author contributions.* FH performed the analyses and wrote the paper. DMS supervised the research, and helped
write and edit the paper.

*Competing interests.* The authors declare that they have no conflicts of interest.

*Acknowledgments.* This article is derived from FH's BSc dissertation at the University of Manchester. We thank
the European Severe Storms Laboratory for their kind access to the ESWD that made this work available. We
thank Neil Mitchell for his comments on the dissertation. We thank Igor Laskowski for explaining the Polish hail
dataset. We thank the anonymous reviewers for their comments that have improved this article.

*Financial support.* Schultz is partially supported by the Natural Environment Research Council grant numbers
NE/N003918/1 and NE/W000997/1.

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
