# Peer review of "Climatology of Large Hail in Europe: Characteristics of 2 the European Severe Weather Database"

_EGUsphere, 2023_

## Author Response (AR1)

1)

The authors present a well-written, clearly structured paper on the usability of ESWD data with respect to hail events. Complementing previous publications, the question is addressed whether an extension of the data base over a longer period of time and to data with a lower quality level leads to comparable results. The core statement that a homogeneous basis for hail climatology has only been achieved over the last 20 years is clearly shown. The paper convinces with the evaluation regarding the relation of hail days and hail events, which can be well used to analyze national and temporal inhomogeneities in the hail reports of ESWD, which is well shown by the hail data from Poland before 1955.

Thank you for your kind words on our manuscript. Also, thank you for taking the time to provide comments that will improve our manuscript.

One comment is directed to the better elaboration of this result in Section 7. It would be desirable if the data from Poland between 1947 and 1955 could be more clearly contrasted with those of more recent reports, in order to discuss indications of data inhomogeneities and doubtful entries for later investigations. In particular, these are figures of the seasonal distribution of hail events, the distribution of maximum hail diameter, and the time of day of the reports. In addition, the discussion of the results may be more open: While climatological trends may be considered as a possible but unlikely reason, there is no mention of the possibility of spurious entries that nevertheless meet the quality criteria. Data originating in official documents such as weather service yearbooks have a higher probability of receiving a higher quality level than reports originating from other sources. In the case in question, the data originated exclusively from the Stalin era, during which the Polish agricultural industry faced repression if it failed to meet ambitious agricultural aims. While the entries during this period provide little information on the weather events themselves, they almost exclusively contain very accurate estimates of losses for agricultural products. Although this investigation is beyond the scope of this paper, an open-ended discussion in this direction would be desirable with the caveat that historical data must be critically evaluated.

Thank you for this information.  Are there any citations to this information?  We have added more information about this data from Igor Laskowski.  Specifically:

    We know the addition of this data in the ESWD was due to Igor Laskowski who reports: "those reports were based on annual records collected by a Polish National Institute of Meteorology founded in 1919, now Institute of Meteorology and Hydrology - National Research Institute (https://imgw.pl/instytut/historia). The data was collected via hail questionnaires, which provided information on the size of the hail (vetch-sized, pea-sized, broad bean-sized, hazelnut-sized, walnut-sized, pigeon egg-sized, hen egg-sized and goose egg-sized) and also details about time of its occurrence, storm direction and the size of the expected yield decrease (in percent). The questionnaires were filled in both by agricultural correspondents of the Polish Central Statistical Office (whose number was growing larger, especially in the [19]50s) and existing insurance companies which provided hail insurance at this time. Those records also contain observations of hail reported by observers at meteorological stations."  At the time of this study, data from yearbooks from 1930–1937 and 1946–1955 had been added.

This information has been added L452.

Apart from that, there are only a few minor comments. In section 7 results of the paper on hail reports at Polish weather stations are compared with those of the present paper. However, it is not written whether the criteria for large hail are the same. Thus, together with the large differences in the hail climatology, the question arises whether large hail is compared with sleet here. This possibility should be discussed.

The size of the hail in Poland is reported, and only large hail is included in this climatology. Because there is the possibility of winter ice being included, we added this sentence to the manuscript: "Although these results may indicate a cool-season preference for hail, there is the possibility that ice pellets or graupel might have been classified as hail (e.g., Punge and Kunz 2018)."

This information has been added L469.

Line 36: First supercell on July 27 and second one on July 28? Please check in ESWD

We cite only 28 July as two supercells formed within a small vicinity and which both produced hailstones of 10+ cm. In contrast, the storm on 27 July produced hailstones of only 7.5 cm. Although much damage was caused by both these events, we only cite one for conciseness. Readers may refer to the cited paper to learn more about this specific event. No change to the manuscript.

Lind 85: In recent years, ESSL also developed an mobile phone app to report severe weather (EWOB)

Yes, we have added a new sentence: "Since December 2015, reports have also been collected via ESSL's European Weather Observer app (Groenemeijer et al. 2017)."

This sentence has been added L 104.

Line 165: The regression lines are not visible in figure 2.

Thank you. We have added those regression lines in the revised manuscript.

The regression lines have been added, now L192.

Line 169: Numbers with plural -s instead of "number"?

Here, we refer to the period of May–July as a single period (as we do with the cool-season months of October–March). So, singular is the correct form. No change to the manuscript.

Figure 7: You may consider to extent the data basis to years before 2000 to show the time of stabilization mentioned in lines 239 and 240.

In fact, only a few figures in the manuscript (Figs. 1, 2, 12, and 13) show data before 2000. The reason for choosing to display the data since 2000 is clear from Figure 1. The number of reports increases substantially after 2000. Thus, unless our purpose was to show the time series of Poland data over a longer period in Figs. 12 and 13 because that is the period of interest, we prefer to limit our graphics to those with the most data (i.e., since 2000). No change to the manuscript.

Line 249: "suggests that the entries that the average hail size is" Delete "that the entries" or "that the average".

Agreed. This has been amended.

Deleted "that the entries", see L326.

Figure 8: "Two red dots represent likely data-entry errors": The red dots are not red in the figure.

We have changed the figure caption to "pink".

We have changed the word to pink on what is now Figure 9 (L332).

Line 302: "would imply" instead of "would implies"?

Yes, amended.

Amended, see L393.

Lines 302 and 303: "except for Germany which has a much greater number of reports proportional to the number of days." And Poland as well according to the graph?

Yes, we agree. We have added "and Poland".

This has been added L 394.

2)

Review Climatology of Large Hail in Europe: Characteristics of the European Severe Weather Database

The paper presents a statistical analysis of large hail reports from the ESWD for Europe. Analyses performed for 120- and 20-year periods include time series of reports and hail days, diurnal and seasonal cycles, annual distributions of hail sizes, and trends in temporal accuracy. Additional emphasis is given to reports from Poland, which has very high numbers of reports in some 10-year periods since 1930.

In general, the paper is well written and clearly structured.

Thank you for your kind words on our manuscript. We appreciate the time that you spent to provide these comments to improve our manuscript.

However, I have some **major concerns**, mainly about the quality and reliability of the data the analyses are based on, and the scientific content.

1. It is difficult for me to see new scientific results and profound conclusions that provide new insights into hail statistics. The paper is of course nice to read, but the scientific value seems to be low.

   We disagree with this assessment. This manuscript analyses more European hail reports than any other study. The closest comparator contains 39,537 reports for a 13-year period. We show the value of QC0+ data in the ESWD. We also examine how these reports have changed over time. The manuscript also documents the large addition of Polish data in the last century. All of these contributions justify the publication of this work.

   Also, the conclusion section is more or less a summary rather than a presentation of conclusions and interpretations.

Indeed, we wrote the conclusion to summarize the manuscript. This approach is an entirely acceptable method for concluding a scientific manuscript. According to the instructions of the journal (https://www.natural-hazards-and-earth-system-sciences.net/submission.html), all that is specified is that each submission contain a conclusions section. At least some scholars of the scientific communication process argue that a conclusions section should contain no new information (e.g., Geerts 1999; Schultz 2009, p. 44) or allow for the possibility of the type of conclusion written here (e.g., Glasman-Deal 2021, p. 245). No change to the manuscript.

Geerts, B., 1999: Trends in atmospheric science journals: A reader's perspective. *Bull. Amer. Meteor. Soc.,* **80,** 639–651.

Glasman-Deal, H., 2021: *Science Research Writing for Native and Non-native Speakers of English.* World Scientific, 356 pp.

Schultz, D. M., 2009: *Eloquent Science: A Practical Guide to Becoming a Better Writer, Speaker, and Atmospheric Scientist.* American Meteorological Society, 412 pp., https://doi.org/10.1007/978-1-935704-03-4.

2. As far as I understood from the manuscript, the authors considered all ESWD reports in their analyses, irrespective of multiple reports from a single storm, or the country or region affected. Given the large differences of prevailing reports among European countries (as shown in the Table), it can be assumed that the results are dominated by individual countries (e.g. Germany, Russia, Poland), leading to large uncertainties in all estimated quantities.

We are not exactly sure what the reviewer's point is here, so if we are mistaken, we apologize. Indeed, some countries will have a greater number of reports per storm than others, irrespective of urban-reporting and daytime-reporting biases. This has previously been identified by several authors (e.g., Groenemeijer and Kühne 2014; Punge and Kunz 2016; Antonescu et al. 2017; Púčik et al. 2019) and has been alluded to in the manuscript. We will, however, make this point clearer in section 2 to fully communicate the biases present in the dataset.

The following has been added to section 2 (L145):

"The existing ESWD dataset is a result of both meteorological variations in hail and reporting issues, much as other severe-weather datasets have (e.g., Groenemeijer and Kühne 2014; Punge and Kunz 2016; Antonescu et al. 2017; Púčik et al. 2019). Indeed, underreporting from rural areas and nighttime storms may influence this dataset. These and other characteristics of the large-hail dataset will be explored in subsequent sections."

3. In the same sense, hail reports are biased towards daytime and towards larger cities. This effect is difficult to estimate, but at least a profound statement is required (although a spatial analysis with respect to the distance of reports to larger areas may help to get an estimate of the latter effect).

See our response to the previous comment.

4. Is a hail day one with at least one report across Europe (that would make no sense), or have you considered some threshold? For example, is a day with only one 2 cm report considered the same as a day with thousands of reports and hailstones larger than 10 cm? That would be strange.

This study aims to look at the ESWD as a whole. Therefore, yes, the entirety of Europe has been considered. Indeed, one hail day is one where at least one hail observation of 2+ cm has been reported. The concept of a hail day is similar to that of a lightning day or tornado day, concepts that are well accepted in the severe weather community. In principle, hail days should be more robust to these fluctuations in reporting individual hail reports, which is why we have showed this quantity. No change to the manuscript.

Furthermore, it makes no sense to define a hail day for the whole of Europe, with its wide variety of local climates. I would rather suggest limiting it to countries with a high number of reports, for example. I would also suggest considering different thresholds for both hail size and number of reports.

The point here is not to look at individual days, but to put this in perspective on the continent over a longer time period. In principle, hail days should be more robust to these fluctuations in reporting individual hail reports, which is why we have showed this quantity. Thus, the hail-day concept is exactly intending to solve the problem the reviewer identified. The reviewer's proposed solution makes a rather simple concept of a hail day into a much more complicated matter. Furthermore, the reviewer's solution of concentrating on countries with higher reports would not remove any variability within climates. No change to the manuscript.

5. Point 4 also refers to the other analyses, such as the annual and diurnal cycles. It is mentioned that Púcik et al. (2019) divided the study area into at least two parts due to the different climates. Why did you not follow this?

Although we agree that Europe encompasses many climates, how to divide these up can occur in numerous ways. We chose not to classify different climatological zones, in part because of this ambiguity and in part because this was beyond the scope of the research. For example, one may choose to differentiate between a more maritime or more continental climate, but these may then contain other factors such as mountain ranges or plains. Hence, we decided to stick to a general overview of the reported distribution of large hail in Europe. No change to the manuscript.

6. A climatological period is usually defined as 30 years or more. It also includes spatial analysis. Neither is the case in this paper. Therefore, I suggest changing both the title and the wording in the manuscript.

From the *Glossary of Meteorology*, *climatology* is defined as "The description and scientific study of climate. Descriptive climatology deals with the observed geographic or temporal distribution of meteorological observations over a specified period of time. Those climatological data can be averaged over 30 years to produce climatological standard normals." (https://glossary.ametsoc.org/wiki/Climatology). Thus, a 30-year period is only relevant for defining climate normals and is not a factor with climatologies of weather events. Also, the geographic distribution is not required for a climatology. Thus, our study fits perfectly with the accepted definition of a climatology.

Furthermore, *NHESS* commonly publishes climatologies of weather events that are not 30-year periods.

**18 years:** Gatzen, C. P., Fink, A. H., Schultz, D. M., and Pinto, J. G.: An 18-year climatology of derechos in Germany, Nat. Hazards Earth Syst. Sci., 20, 1335–1351, https://doi.org/10.5194/nhess-20-1335-2020, 2020.

**15 years:** Burcea, S., Cică, R., and Bojariu, R.: Radar-derived convective storms' climatology for the Prut River basin: 2003–2017, Nat. Hazards Earth Syst. Sci., 19, 1305–1318, https://doi.org/10.5194/nhess-19-1305-2019, 2019.

**10 years:** Pacey, G., Pfahl, S., Schielicke, L., and Wapler, K.: The climatology and nature of warm-season convective cells in cold-frontal environments over Germany, Nat. Hazards Earth Syst. Sci. Discuss. [preprint], https://doi.org/10.5194/nhess-2023-39, in review, 2023.

**10 years:** Akkoyunlu, B. O., Baltaci, H., and Tayanc, M.: Atmospheric conditions of extreme precipitation events in western Turkey for the period 2006–2015, Nat. Hazards Earth Syst. Sci., 19, 107–119, https://doi.org/10.5194/nhess-19-107-2019, 2019.

For these reasons, we disagree with the premise of the comment. No change to the manuscript.

Additional **minor review points** are those:

1.  L21: 20-years is not a climatological period (major comment 5)

    We disagree. See our response to the previous comment. No change to the manuscript.

2.  L30: "Large hail" for a diameter of > 2 cm is not a European definition, rather used by ESWD.

    Thank you. Deleted "in Europe".

    Added, see L30.

3.  L44-45: You may add that most of the hail climatologies / statistics (e.g., those cited in Touvinen et al., 2009) are outdated

    What the reviewer means by "outdated" is unclear and unfair to these studies. Indeed, some of the studies mentioned were published a number of years ago as implied by our statement of ""A summary of *past* European hail climatologies". However, this does not mean their results are necessarily outdated. Moreover, readers would understand that a study published in 2009 is representative of the time in which it was published and of the dataset from which it was derived. Therefore, we disagree with the premise of this comment. No change to the manuscript.

4.  L50: It should be noted here that some pan-European hail hazard assessments are available, e.g. from Punge et al. 2014 or Punge et al. 2017 based on overshooting top detections, from Rädler et al. 2018 using reanalysis, or from Taszarek et al. 2018 using multiple data sources. In this sense, the statement in L60 "…their work shed the first light on" is not true.

Thank you for this clarification. Indeed, previous climatologies do exist, but are not based on the ESWD data as a singular entity. We have clarified this by putting "from surface reports" at the end of the sentence.

Moreover, comparing these other climatologies with our results will be discussed in the results sections, per a comment by Reviewer 3. We believe that these revisions will also address this present comment.

The following has been added to section as a response to this comment (L584):

"Previous studies have provided pan-European climatologies of hail based using other methods such as Punge et al. (2014, 2017) who used overshooting cloud tops, Rädler et al. (2018) who used reanalysis data, or Taszarek et al. (2018) who used a combination of data sources. Some studies are projecting increases in hailstorms with climate change in Italy (Piani et al. 2005), Netherlands (Botzen et al. 2010), and Germany (Mohr et al. 2015), as well as across much of Europe (Taszarek et al. 2021). Other studies have also concluded that there were no positive trends in the frequency of hail in hailpad data in northern Italy and France (e.g., Eccel et al. 2012; Dessens et al. 2015; Raupach et al. 2021; Manzato et al. 2023). Tazarek et al. (2019) argue that a combination of datasets is important to construct a robust climatology, particularly as the spatial and temporal resolutions would often differ between methods. Furthermore, studies such as Rädler et al. (2018) compared their reanalysis results to surface observed reports from the ESWD to strengthen their arguments. Therefore, understanding the characteristics of the current surface observations via the ESWD helps not only build a climatology of large hail in Europe, but can also be used in association with other research methods to identify the underlying factors which lead to such events."

5. I miss a better motivation and scientific objectives of the paper. "Increasing the size of the dataset through…extending the period of analysis" is too weak when only 2 additional years are considered.

This comment is unfair. The reviewer has selectively edited this sentence to misrepresent what we actually wrote in the original submission.

"In the present article, we explore whether increasing the size of the dataset through *lowering the quality-control levels of the reports* and extending the period of analysis yields comparable results, increasing the generality of Púčik et al.'s (2019) results."

So, our analysis was also about adding cases through lowering the quality-control levels of the reports, not only extending the time period. These two changes resulted in an increase in the number of reports from 39,537 (Púčik et al. 2019) to 62,053 (present study), a 57% increase in the size of the dataset.

But, our study is about more than just increasing the size of the dataset. We also had different purposes to Púčik et al. (2019), which again were not mentioned by the reviewer.

"In doing so, we also document the reporting characteristics of the database as a function of time both throughout the 20th century and within the last 20 years. In particular, we seek the possible existence of a relatively homogeneous period of time in the database that could be used as a baseline for climatologies and climate-change studies."

Thus, we feel that we have clearly stated our motivation and scientific objectives, despite the manipulated and truncated quotation provided by the reviewer. No change to the manuscript.

6. P3, 2ⁿᵈ paragraph: Why did you not use the most recent data until 2022? The analyses seem to be easily reproducible.

This manuscript was a result of an undergraduate dissertation (see the acknowledgements). The study commenced in late 2020. The data was sent from the ESWD, so the dataset was set as of late September 2020. Further analysis was not necessary. Nevertheless, we have added "at the time this study commenced" in section 2 to make it clear to other readers that the scope of the dataset was determined at this time.

The sentence "At the time this study commenced" has been added (L106).

L98-105: This is the correct designation of the quality levels; in the later text they are incorrectly quoted.; L106: "…plausibly checked QC1…", but this is "report confirmed"

Deleted "plausibly checked". Thank you.

Amended, see L126.

7. Also in L225-226 it should read "report confirmed"

Added "confirmed". Thank you.

Amended, see L293.

8. L157: "…ability to detect reports linked to the same event, and hence have removed duplicate events from the dataset". This would make no sense at all and is not the case. In the papers cited (e.g. Wilhelm et al., 2020) it is clear that a single streak is covered by several reports.

The point made here is that fewer reports have been needed for the same quantity of hail days over recent years than previously. Therefore, we are just speculating a few reasons for this. No change to the manuscript.

Just a small correction to note in this comment: The citation should be Wilhelm et al. (2021), not (2020).

9. L77-79: Kunz et al. (2020) estimated annual and diurnal cycles not from ESWD data, but from radar-derived potential hail streaks (Z > 55 dBZ). These streaks were also combined with ESWD reports. The main difference is not the quality level of the ESWD reports considered because as written in Sect. 2, 70.4% were QC1 and 29% were QC+, leaving only 0.6% at Q0 level.

We presume this refers to lines 177–179 where we cite Kunz et al. (2020), not lines 77–79.

Thank you for this clarification. We have revised the sentence to the following:

"These distributions are also similar to those from Kunz et al. (2020, their Fig. 2a) for hailstorms in central Europe using radar-derived hail streaks combined with all quality

levels from the ESWD, indicating that this dataset derived using different methods is a reliable source of large-hail data."

This sentence has been changed to "These distributions are also similar to those from Kunz et al. (2020, their Fig. 2a) for hailstorms in central Europe using radar-derived hail streaks combined with all quality levels from the ESWD, indicating that this dataset derived using different methods is a reliable source of large-hail data." And is now found at L 207.

10. L188: Can you briefly describe how you converted UTC to LT?

All reports have the country of origin listed and all times are in UTC. By looking at each country on an individual basis, these were converted to LT taking daylight savings into account. No change to the manuscript.

11. L191-192: see comment (9); Although the diurnal cycles of Kunz et al. (2020) have a resolution of only 3 hours, there are some differences, which may be due to different study areas?

In fact, Figure 4 (local time) in the present manuscript if converted to a bar chart and Fig. 2b in Kunz et al. (2020) are quite similar. Sure, small differences will be due to different study areas and different years, but we don't see that. No change to the manuscript.

12. Fig 7: This figure is very interesting, but again not very valuable for the whole of Europe (and the under-reporting in most countries). I suggest that this type of figure be reproduced for countries where the number of reports is highest according to the Table.

We find that this figure remains interesting by showing that there is not that much variation in the peak hail time across Europe, even between different climatic zones and countries. However, we do see the value in adding a table showing the proportion of hail days per year by country. We also believe that a figure showing the annual distribution of hail reports per country could be interesting, as a more even spread would suggest more consistent reporting over the years.

We have added the following in response to this comment:

**L219:** The percentage of hail days reported by month per country for the period 2000–2020 was investigated in Fig. 4. Greece is the only country to not have over 50% of its reports being within the months of May, June, and July, having a more consistent number of hail days throughout the year. Many countries do not have any reports before April or after September. Spain, Italy, France, and Croatia have similar distributions of hail days throughout the year, which may be linked to their Mediterranean setting, although Slovenia, Bosnia and Herzegovina, and Bulgaria do not share the same characteristics, despite also being situated along the Mediterranean. Previous studies such as Tazarek et al. (2020) have investigated hail distribution in Europe by linking events to meteorological and climatological factors, which may help explain some of the differences seen in Fig. 4. Furthermore, Sanchez et al. (2017) investigated hail events in southern Europe, concluding that even small geographical and climatological differences can have a large impact on the number of hail days reported, but also with the peak month of hailfall, which may also explain some of the differences in Fig. 4.

[Figure]

**Figure 4. Horizontal bar charts of the monthly distribution for countries with 100 or more reports: 2000–2020.**

**L399:** We further investigated the hail size distribution by country for the period 2000–2020 (Fig. 13). Only one report of each size diameter was taken per country per day to minimize some of the reporting biases. Finland has the greatest proportion of the lowest hail bin size, whereas Slovenia has the lowest. For sizes 5 cm in diameter and greater, the proportion of hail sizes recorded starts to diminish drastically, which would be expected as larger hailstones are rarer. Although Slovenia has the greatest proportion of hail sizes above 5 cm, these reports came from a sample of 116 hail reports, one of the smallest of the countries analyzed. For hail days with a report above 10 cm, Russia has the greatest quantity with 10 reports over this period, whereas Italy came second with 9 reports and France with 8. Slovenia, although having a greater proportion, had 5 days with a hail report above 10 cm for this period.

[Figure]

**Figure 13. Horizontal bar charts of the size distribution of large hail for countries with 100 or more reports: 2000–2020.**

L248 and L300: Did you use the Pearson product-moment or Spearman rank correlation coefficient? The latter would be more appropriate due to the obvious deviation from a normal distribution.

We used the Pearson product-moment. However, the reviewer is correct and the Spearman correlation would make more sense. We will amend this.

This has been amended. L396 Figure 12.

13. L287-289: The main reason for the high number of reports in Germany is obviously that ESSL was founded here.

Yes and no. The ESWD grew out of other data-collecting efforts such as TorDACH (tornado dataset from Germany, Austria, and Switzerland). So, although there was a focus on Germany, it was not strictly limited to the founding of ESSL. No change.

It should be mentioned that in some countries severe weather reports are collected by other institutions, e.g. KERAUNOS in France. Moreover, crowd-sourcing via meteo apps is well known and emerging in some countries, such as the MeteoSwiss app, which has collected >100,000 reports in recent years (compared to only 266 ESWD reports). So we should not blame spotters for being less enthusiastic.

The wording as written is precise. There are two factors in play here, and our text is clear in both of those factors. Storm-spotter networks may be more or less enthusiastic about collecting reports within their own countries ("existence, size, and enthusiasm of spotter networks within each country"), and such networks may vary in how effective they are at contributing those reports to the ESWD ("variations in the ability or enthusiasm of citizens to input into the ESWD"). No change to the manuscript.

14. L315 and others: I'm not sure about the comparability with the study by Suwala (2011), as they used station data over a period of 8 years. Station data often do not distinguish between hail diameters, but rather consider ice pellets or graupel in the same class as hail (see also the review by Punge and Kunz, 2018). This could at least explain the discrepancies with the large number of hail events in the cold season. However, this is frankly speculative, as there is no information available for the Polish station.

This is a fair point. We have now added some text to clarify this point. "Although these results may indicate a cool-season preference for hail, there is the possibility that ice pellets or graupel might have been classified as hail (e.g., Punge and Kunz 2018)." Thank you for this information.

This sentence has been added, see L469.

15. L324: Again, a separation by region would be desirable.

Please see our response to point 5.

16. L374-377: The trend directions are not that clear. Eccel et al. (2012) or Manzato et al. (2023) found no positive trends in hailpad data in northern Italy, but fewer and larger hailstones. Dessens et al. (2015) found almost the same in their hailpad data in France. You may cite here the review paper by Raupach et al. (2021).

Thank you. We have added a sentence to the manuscript. "Other studies have also concluded that there were no positive trends in the frequency of hail in hailpad data in

northern Italy (e.g., Eccel et al. 2012; Dessens et al. 2015; Raupach et al. 2021; Manzato et al. 2023)."

Section 8 has been amended to contain the following, encompassing our response to this comment:

"Previous studies have provided pan-European climatologies of hail based using other methods such as Punge et al. (2014, 2017) who used overshooting cloud tops, Rädler et al. (2018) who used reanalysis data, or Taszarek et al. (2018) who used a combination of data sources. Some studies are projecting increases in hailstorms with climate change in Italy (Piani et al. 2005), Netherlands (Botzen et al. 2010), and Germany (Mohr et al. 2015), as well as across much of Europe (Taszarek et al. 2021). Other studies have also concluded that there were no positive trends in the frequency of hail in hailpad data in northern Italy and France (e.g., Eccel et al. 2012; Dessens et al. 2015; Raupach et al. 2021; Manzato et al. 2023). Tazarek et al. (2019) argue that a combination of datasets is important to construct a robust climatology, particularly as the spatial and temporal resolutions would often differ between methods. Furthermore, studies such as Rädler et al. (2018) compared their reanalysis results to surface observed reports from the ESWD to strengthen their arguments. Therefore, understanding the characteristics of the current surface observations via the ESWD helps not only build a climatology of large hail in Europe, but can also be used in association with other research methods to identify the underlying factors which lead to such events."

**Edits/Typos**:

L18: "…dataset for severe convective storms **reports**." Otherwise it's not true, as there are SCS statistics available from model data or overshooting top reports (see minor point X)

Fixed. Thank you.

See L18.

L20: "…to evaluate **hail reports** from… " (you did not evaluate the database)

Fixed. Thank you.

See L20.

L38: "on 10 June"; you refer here to the supercell that hit the city of Munich on that day.

Now reworded to: "Other similar events occurred over southern Germany on 10–12 June 2019, with one storm producing 6-cm hailstones and causing EUR 1 billion in damages".

This has been amended, see L38.

L43: "…intensity, and hailstone size." On L23, you wrote "intensity, as measured by maximum hail size..", but here you both intensity and size.

"Intensity" has been deleted.

See L43.

L54: "..which help**s**.."

Fixed. Thank you.

See L67.

L83: delete "insurance data information": in the ESWD data, I see only 3 entries in 20 years that are from an insurance company; this is not worth mentioned here

So, it is not a lot of information, but it is present in the ESWD. So, our statement is correct. No change to the manuscript.

L84: delete "organizations"; a large number of reports are not from organizations rather than from trained (and well-known) spotters

Fixed. Thank you.

See L103.

L93 "…also examin**ed**…" above and below you used past tense

Fixed. Thank you.

See L113.

L215: "…frequency of **events**…" for one event, the frequency is = 1;

Fixed. Thank you.

See L283.

L218: "…**is** more spherical…"

We disagree. "Were" is being used in the past subjunctive tense and is being used correctly. No change to the manuscript.

L224: decreased --> decreases

We disagree. This verb is best as past tense, consistent with our interpretation of the data in the figures, which occurred in the past. No change to the manuscript.

L273: delete from

Fixed. Thank you.

See L357.

L298: include a comma after Fig. 2

We do not agree that a comma is needed here. No change to the manuscript.

L302: implies --> imply

Fixed. Thank you.

See L393.

L330: mentions --> mentioned

Fixed. Thank you.

See L480.

3)

This study explores some characteristics of hail in Europe based on hail reports from ESWD. A distinction is made between hail days and hail reports. The data are summarized in graphics. The seasonality and diurnal cycle are described and data with two different quality flags compared. The time evolution of the reported hail stone sizes is illustrated.

Thank you for these comments. They will help us improve the manuscript.

One major point that I would like to raise is that the paper does provide only very limited information about the data in the data base, the quality check procedures, and the methodology that is used to analyse the data. This information is crucial for the interpretation of the data and the discussion of the limitation and uncertainties of the data set (more detailed information on that point is listed below).

We agree that some of that information is needed to understand this study. Where we felt that information was needed or relevant to our arguments or results, we have included that information. However, there are many other documents that contain the procedures and dataset information in much more detail than we can provide. For more information on the functioning of the ESWD, please refer to any of these documents: Groenemeijer et al. (2009, 2017), Dotzek et al. (2009), and Groenemeijer and Liang (2020).

The second point is that the analyses are qualitative. This is ok but there should be no statements about changes over time in the abstract without underlying statistical analyses.

We certainly appreciate the importance of quantitative analyses where relevant and possible. But, we disagree with the rigid statement "no statements about changes over time...without underlying statistical analyses". Certainly, some signals can be so strong or so weak that statistical analyses are not needed to confirm what is visually apparent, especially with over 60,000 hail reports. We address each statement individually in the comments below.

Major:

- Please provide (a lot) more information about the data sources of ESWD. What are all the data sources of ESWD (e.g. does it also contain insurance data?)? How big is

the fraction of each data source (e.g. crowed-sourced vs. observers from weather services)? How do the different data sources change over time? This information is important as the uncertainty crowd-sourced data and insurance data is quite high

How large is the fraction of each data source in each quality class? How exactly are the quality classes assigned? What does plausibility checked mean exactly? A cross-check against radar information? A cross check against newspaper reports? Which data source provides typically, which types of variables (e.g. mean and max. size of the hail stones).

Please include all of this information in the methods section.

This level of detail is tedious and unnecessary for most of what we are doing. Other published articles that have used the ESWD have not had to answer these types of questions to the same level of detail. These published articles do not contain this information, even if such answers were available.

We agree that some of that information is needed to understand this study. Where we felt that information was needed or relevant to our arguments or results, we have included that information. However, there are many other documents that contain the procedures and dataset information in much more detail than we can provide. For more information on the functioning of the ESWD, please refer to any of these documents: Groenemeijer et al. (2009, 2017), Dotzek et al. (2009), and Groenemeijer and Liang (2020).

Hail days: Please provide more information in the methods section how you identify hail reports and hail days?

The reports come directly from the ESWD as individual records in a spreadsheet. Therefore, we do not 'identify' these ourselves. How hail days are computed is already described in the text: "The annual number of large-hail days was derived from the annual number of large-hail reports by removing duplicate dates." We just counted up the number of unique dates in the dataset to obtain the number of hail days. No change to the manuscript.

How can you have more hail days than hail reports (Figure 1)?

There are never more hail days than hail reports per year. The scales are on different axes: left *y* axis for reports and the right *y* axis for days. No change to the manuscript.

Hail events: Please explain how you remove duplicate dates. How do you define an event? What counts as a duplicate date? If the report is exactly at the same location? In the same country? How much time difference do you allow for? How accurate are the report locations?

Duplicate dates are simply when more than one hail event is recorded across Europe on the same day, regardless of location. When we considered hail days per country, then the same procedure applies.

Time accuracy: If I understand it right, this information is self-declared? Has it every been verified against independent data (radar, satellite information)? How is this information obtained for historical data? Please expand the discussion of this variable to include these aspects in the methods section. How reliable is time information generally in crowd-sourced data?

This information is added by the ESWD data manager at ESSL. It is not self-reported. It is unclear how to verify this value as we never have the true time of the event down to the second. The time accuracy quantity is just a statement of how precise the reporting time of

the event is.  For example, if the newspaper story and photo that confirms the hail event says 15 local time, then the time accuracy is one hour. That is all.  No changes to the manuscript.

Location accuracy: how is this parameter estimated? How is it verified?

Same as the previous comment. No changes to the manuscript.

2) How do your findings compare to hail climatologies based on radar/satellite and proxy indicators?

This is a great suggestion.  Comparing our results to those of previous studies are further discussed in section 8, among other places in the text where they are most appropriate.

We believe that the following papers would be relevant to our findings.

Sanchez, J.L., Merino, A., Melcón, P., García-Ortega, E., Fernández-González, S., Berthet, C. and Dessens, J., 2017. Are meteorological conditions favoring hail precipitation change in Southern Europe? Analysis of the period 1948–2015. *Atmospheric Research*, *198*, pp.1-10.

This paper discusses the variability in hail days from different climates in France and Spain over the time period 1948–2015. They highlight that only small spatial variations have a large impact on the number of hail days recorded over the time period. Furthermore, the paper emphasises that different climates have different peak hail months within the year. Therefore, we believe that an annual distribution chart per country could also be an interesting addition to the manuscript.

Rädler, A.T., Groenemeijer, P., Faust, E. and Sausen, R., 2018. Detecting severe weather trends using an additive regressive convective hazard model (AR-CHaMo). *Journal of Applied Meteorology and Climatology*, *57*(3), pp.569-587.

This study uses reanalysis data from 1979 onwards, but also compared them to ESWD reports. This highlights that observed events play an important role even for other methods of researching severe events to ensure that results are plausible.

Taszarek, M., Allen, J., Púčik, T., Groenemeijer, P., Czernecki, B., Kolendowicz, L., Lagouvardos, K., Kotroni, V. and Schulz, W., 2019. A climatology of thunderstorms across Europe from a synthesis of multiple data sources. *Journal of Climate*, *32*(6), pp.1813-1837.

This study looks at different datasets in order to investigate severe storms in Europe. They argue that by using different datasets, and therefore data collection methods, you can compare the spatial and temporal resolutions of these. This highlights the importance of understanding the strengths and weakness of different datasets, as well as understanding their composition, which our study provides from the ESWD at a lower quality-control level than previously published.

Taszarek, M., Brooks, H.E., Czernecki, B., Szuster, P. and Fortuniak, K., 2018. Climatological aspects of convective parameters over Europe: A comparison of ERA-Interim and sounding data. *Journal of Climate*, *31*(11), pp.4281-4308.

We also believe that this study could make an interesting comparison, as it investigates annual distribution by region, with respect to the underlying meteorological factors that provide ideal environments for severe storm production.

The following has been added in response to this comment:

**L59:** Climatologies of European convective storms and their impacts have been constructed using a number of datasets. For example, some studies have examined the climatology of convective storms using remote-sensed data such as lightning, radar, and satellite (e.g., Punge et al. 2017). Others have examined the environments that favor such storms, such as through reanalyses or soundings (Rädler et al. 2018; Taszarek et al. 2017, 2018, 2019) or reanalyses coupled with hailpad data (Sanchez et al. 2017).

**L70:** For example, Taszarek et al. (2019) found substantial variability across Europe in the frequency of ESWD reports and the frequency of favorable environments for convective storms.

**L224:** Previous studies such as Tazarek et al. (2020) have investigated hail distribution in Europe by linking events to meteorological and climatological factors, which may help explain some of the differences seen in Fig. 4. Furthermore, Sanchez et al. (2017) investigated hail events in southern Europe, concluding that even small geographical and climatological differences can have a large impact on the number of hail days reported, but also with the peak month of hailfall, which may also explain some of the differences in Fig. 4.

**L584:** Previous studies have provided pan-European climatologies of hail based using other methods such as Punge et al. (2014, 2017) who used overshooting cloud tops, Rädler et al. (2018) who used reanalysis data, or Taszarek et al. (2018) who used a combination of data sources. Some studies are projecting increases in hailstorms with climate change in Italy (Piani et al. 2005), Netherlands (Botzen et al. 2010), and Germany (Mohr et al. 2015), as well as across much of Europe (Taszarek et al. 2021). Other studies have also concluded that there were no positive trends in the frequency of hail in hailpad data in northern Italy and France (e.g., Eccel et al. 2012; Dessens et al. 2015; Raupach et al. 2021; Manzato et al. 2023). Tazarek et al. (2019) argue that a combination of datasets is important to construct a robust climatology, particularly as the spatial and temporal resolutions would often differ between methods. Furthermore, studies such as Rädler et al. (2018) compared their reanalysis results to surface observed reports from the ESWD to strengthen their arguments. Therefore, understanding the characteristics of the current surface observations via the ESWD helps not only build a climatology of large hail in Europe, but can also be used in association with other research methods to identify the underlying factors which lead to such events.

Minor points:

Abstract: The instensity as measued by …. ◊ larger hailstones are rarer than smaller hailstones.

Of course they are. Nevertheless, the maximum hail size is often the measure of the intensity of a hailstorm, as commonly accepted by the meteorological community. No change to the manuscript.

Introduction: there is a body of literature that discusses hail climatologies in Euope based on indirect observations (radar, satellite data) and proxy indicators (soundings, renalaysis). I recommend a qualitative comparison of the results with this body of literature and hence a brief discussion of this research branch in the introduction.

Good suggestion. The third paragraph of the introduction discusses those past climatologies. The comparison to the results of the present study are further discussed throughout the text where they are most appropriate and in section 8. See suggested studies in point 2.

The following has been added in response to this comment: and point 2

   **L59:** Climatologies of European convective storms and their impacts have been constructed using a number of datasets.  For example, some studies have examined the climatology of convective storms using remote-sensed data such as lightning, radar, and satellite (e.g., Punge et al. 2017).  Others have examined the environments that favor such storms, such as through reanalyses or soundings (Rädler et al. 2018; Taszarek et al. 2017, 2018, 2019) or reanalyses coupled with hailpad data (Sanchez et al. 2017).

L86 please add: at the time of this study …

Yes, fixed.

This has been amended, now L106.

L106 please mention if these levels are inclusive, i.e. are all QC1 also included in Q0+?

No, these are separate categories.  Each report only has one category. We have revised the manuscript to be more clear, referring to a "single" quality-control level and being explicit about which categories were included: "Púčik et al. (2019) used only plausibly checked QC1 and QC2 events" and "this present study uses QC0+, QC1, and QC2."

This sentence has been revised in reponse to this comment (L126):

"As mentioned in section 1, Púčik et al. (2019) used only QC1 and QC2 events. However, to see if the quality-control level affects the interpretation of the results, this present study uses QC0+, QC1, and QC2."

L106 please mention that the quality control is strictly against reports from ESWD with a higher quality flag not with other data sources.

Indeed.  Fixed.  Please see our response to the above comment.

L179 "This data set" is referring to QC0+ correct? Please state so explicitly.

Correct.  We have added "this larger dataset including QC0+ events…" to clarify.

This sentence has been amended to "These distributions are also similar to those from Kunz et al. (2020, their Fig. 2a) for hailstorms in central Europe using radar-derived hail streaks combined with all quality levels from the ESWD, indicating that this larger  dataset of QC0+ events derived using different methods is a reliable source of large-hail data."

L205

No comment was provided here, just a line number.  It might be that there was a missing "and" in this sentence.  We have fixed that.  Thank you.

L225 pausibly ◊ plausibility?

The wrong QC description was given here.  In fact, this report was "confirmed", and we have changed the wording accordingly to "confirmed".  Thank you.

This has been amended, see L293.

L234 does not change dramatically ◊ please compare distributions and check for stat. significance

Figure 7 has very little variability in the data year to year. This fact can be seen visually by readers. If there is no visual indication of a reasonable amount of variability, then there will be no reason to test for statistical significance. No change to the manuscript.

L237 a period of stability ◊ of what?

Stability in reporting. Specifically, on average over this time period, roughly the same percentage of each bin size were reported. We have revised the text to include "in reporting".

This has been amended, see L311.

Figure 7: Can there be a bias towards larger hail stones in crowd-sourced data? People being eager to report large hail stones? How does it compare to ground observations from e.g. hail pads?

The biases in the hail-size reports have been documented in other publications. Indeed, as likely as it is that maximum hail sizes may be overestimated, maximum hail sizes may also be undersampled. All the data in this manuscript (as with all other hail studies using the ESWD) will have the same issues (we prefer the word "issue" rather than "bias"). There are few studies that use direct measurement from hail pads. Even so, it is unclear how such studies could be applied to our study. So, while we do not disagree with the reviewer, we have no way of rectifying these issues. No change to the manuscript.

L276 This statement is a bit patronizing, I recommend removing it.

We have rephrased this sentence to "Figure 10 also indicates the countries for which there is opportunity to improve engagement in severe-weather reporting."

This has been amended, see L360.

L281 suggest to add "People in countries …"

Revision is unnecessary as it refers to countries as a combined effort by spotters and other organizations (including the national hydrometeorological services) in each country, not an evaluation of personal contributions by any one person. No change to the manuscript.

L287 Note that some countries such as Switzerland and Germany have national hail crowd-sourcing programs organized through the National weather services that might explain why there are fewer entries to ESWD

Germany has the most large-hail reports (4956) and Switzerland has among the least. So, it would seem difficult to argue that such programs are influencing the number of reports. We would feel uncomfortable making such an assertion, given our results. No change to the manuscript.

Section 7: It is not entirely clear why you dedicate such a detailed analysis to the data from Poland.

Figure 1 shows some unusual behaviour in the 1930s, 1940s, and 1950s. Further analysis of those unusual maxima in hail reporting results in Figure 12, which shows that these maxima

are entirely a result of Polish data. This is the premise of our argument in the first paragraph in section 7. Not mentioning these unusual maxima would be inappropriate.

However, recognizing that we have a large number of reports from a different century, it would be interesting to compare their characteristics to the characteristics of the modern dataset from the ESWD. This comparison is the reason for Figure 13 (now 15), as well as the rest of the text in section 7. We believe it is a worthwhile inclusion to the manuscript. No change to the manuscript.

L353 the reported location accuracy

Included "reported".

See L555.

L359 I do not yet understand how you come to this conclusion about hail trends.

"Hail trends" could have been better worded for clarity. We have revised these words to "distributions of hail size, frequency, and location".

This has been amended see L561.

L377 consistent = homogeneous? If not, what do you mean exactly by consistent?

"consistent" has been deleted.

L396ff These statements need to be supported by statistical analyses

See our previous argument. No change to the manuscript.

L399 the reported time accuracy

Adding "reported" is unnecessary. "Reports" or "reported" is already used three times in this sentence. The implication is clear. No change to the manuscript.

---

## Author Response (AR2)

**Response to Reviewer's Comments: Round 2**

Reviewer's comments in black.  Authors' original response in red.  Authors' response this round in purple.

—

The revised version of the manuscript shows improvement as the authors have taken into account some of the questions and suggestions raised by the reviewers. However, it is disappointing to note that several important points raised by the reviewers were not adequately addressed or considered by the authors, but rather simply dismissed by stating "no change in the manuscript."

We are sorry that our response has led to the reviewer feeling this way. It was not our intention to dismiss the comments by the reviewer.  We felt that we have tried to understand each comment, consider our response, then argue for a course of action that we were comfortable doing. None of our responses were intended to dismiss the comment without consideration.

That said, we disagree that we did not adequately address the reviewer's comments in our response. From our perspective, we responded professionally and thoroughly to each of the reviewer's comments, justifying our choices and responding to the reviewer's concerns.  We did not understand some comments, we disagreed with others, and we felt that not all comments warranted a change to the manuscript.  That is why we wrote "no change to the manuscript" for a minority of our responses. That statement is commonly used in reviews and was not intended to dismiss the reviewer's comments. It was merely a factual statement stating we felt that our response was sufficient and no further change was needed in the manuscript.

In my review, I emphasize the need for the authors to adequately address and consider the major revision points 4 and 5, as well as the minor points 3, 5, 8, 10, 11, and 13 (second paragraph, although this is not the most critical issue).

In this next round of reviews, we have endeavored to revise the manuscript where possible to address the reviewer's concerns.  We hope you find them acceptable.

**Major Point 4:**

4. Is a hail day one with at least one report across Europe (that would make no sense), or have you considered some threshold? For example, is a day with only one 2 cm report considered the same as a day with thousands of reports and hailstones larger than 10 cm? That would be strange.

> This study aims to look at the ESWD as a whole. Therefore, yes, the entirety of Europe has been considered. Indeed, one hail day is one where at least one hail observation of 2+ cm has been reported. The concept of a hail day is similar to that of a lightning day or tornado day, concepts that are well accepted in the severe weather community. In principle, hail days should be more robust to these fluctuations in reporting individual hail reports, which is why we have showed this quantity. No change to the manuscript.

Furthermore, it makes no sense to define a hail day for the whole of Europe, with its wide variety of local climates. I would rather suggest limiting it to countries with a high number of reports, for example. I would also suggest considering different thresholds for both hail size and number of reports.

The point here is not to look at individual days, but to put this in perspective on the continent over a longer time period. In principle, hail days should be more robust to these fluctuations in reporting individual hail reports, which is why we have showed this quantity. Thus, the hail-day concept is exactly intending to solve the problem the reviewer identified. The reviewer's proposed solution makes a rather simple concept of a hail day into a much more complicated matter. Furthermore, the reviewer's solution of concentrating on countries with higher reports would not remove any variability within climates. No change to the manuscript.

As we have written previously, previous studies on convective phenomena consider the concept of a "day", including over large areas such as Europe (e.g., Pucik et al. 2019). For example, Punge and Kunz (2016) write that hail days are also aligned with information that the insurance industry uses, as their portfolios cover regions larger than countries and hailstorm outbreaks may cover more than one country.

Moreover, why would an arbitrary threshold for a number of reports over a number of locations be needed to be considered a hail day?  Why would that number be any different from the value of 1, when the reports used are plausibility checked?  It is unclear how one would pick a threshold, justify it, and have it be meaningful, in any way that makes more sense than how we have done it, which again is standard in severe-weather climatologies.

Looking at hail days across Europe as a whole for the entirety of the dataset helps to understand the spread of the data over time. This information is then used when considering annual and diurnal cycles, as some countries have more complete records in the ESWD. Hence, this would favor these countries when looking at Europe as a whole. Furthermore, by looking at hail days and reports (Figure 2), more recent years possess a smaller number of hail reports for the same number of hail days. This result can be interpreted as the ESWD as a dataset has gained spatial extent in hail reporting, making it a more valuable resource. Additionally, by comparing hail days and hail reports on an annual cycle (Figure 3), there is a discrepancy in hail reports vs hail days per month. This discrepancy could suggest many things, such as hail being seen more widely across Europe during the early summer season, or an underlying nonmeteorological factor that would affect the number of reports.

For these above reasons, we strongly argue that it makes perfect sense to discuss hail days across the entirety of Europe.  We retain the concept of hail days in the manuscript.  We have added text at lines 124–128 to justify our choice of inclusion of hail days.

"We analyzed not only the number of hail reports, but the number of hail days, as well. Hail days are a more robust measure of hail occurrence and helps minimize variability due to variability in hail reporting across different countries. Hail days are also useful for certain purposes. For example, Punge and Kunz (2016) wrote that hail days are also aligned with information that the insurance industry uses, as their portfolios cover regions larger than countries and hailstorm outbreaks may cover more than one country."

That being said, we appreciate that the European continent contains many different climates. Hence, the climatology of hail may differ among the various countries. Therefore, we have added extra country-by-country information for those countries with 100 or more reports during the period 2000–2020, adding a new Figure 4 that illustrates the annual distribution of hail days by country. We have also added a new Figure 7, which shows the hourly distribution of large-hail reports by country for countries with 100 or more reports. Both figures are supplemented with accompanying text. We hope these changes address your concern.

5. Point 4 also refers to the other analyses, such as the annual and diurnal cycles. It is mentioned that Púcik et al. (2019) divided the study area into at least two parts due to the different climates. Why did you not follow this?

> Although we agree that Europe encompasses many climates, how to divide these up can occur in numerous ways. We chose not to classify different climatological zones, in part because of this ambiguity and in part because this was beyond the scope of the research. For example, one may choose to differentiate between a more maritime or more continental climate, but these may then contain other factors such as mountain ranges or plains. Hence, we decided to stick to a general overview of the reported distribution of large hail in Europe. No change to the manuscript.

In fact, the last revision of the manuscript had included a new figure (then Figure 4) that addressed the variation in monthly distribution across the European countries. That new Figure 4 comprised 24 horizontal bar charts for 24 countries with 100 or more reports. We had also added a new figure (then Figure 13) that addressed the variation in hail-size distribution across the European countries. That new Figure 13 comprised 24 horizontal bar charts for 24 countries with 100 or more reports. A similar plot for diurnal cycle was not performed. We noticed that the reviewer did not comment upon the addition of these figures, which helped address their concerns about the regional variability. Perhaps the reviewer missed these two new figures?

We have added a new figure (Figure 7) that shows the country-by-country distribution of hail reports by hour. We hope this figure and its accompanying text addresses your concern.

We analysed the data initially on a European-wide basis, and then on a country-by-country basis. Several figures in our manuscript address the regional variability of the dataset by country: Figures 4, 11, 12 and 13, and Table 1. Indeed, even within these countries there will be climatic variations. But, the aim of this study was not to construct a detailed intercomparison between individual large-hail climatologies on a country level, but to understand more about the data in the ESWD and to evaluate if similar results can be produced to other studies when considering a lower quality-control level.

We have added a number of new figures in two iterations of this manuscript and new text addressing the reviewers' concerns. We hope these revisions are suitable.

**Minor Point 3:**

> L44-45: You may add that most of the hail climatologies / statistics (e.g., those cited in Touvinen et al., 2009) are outdated

What the reviewer means by "outdated" is unclear and unfair to these studies. Indeed, some of the studies mentioned were published a number of years ago as implied by our statement of ""A summary of past European hail climatologies". However, this does not mean their results are necessarily outdated. Moreover, readers would understand that a study published in 2009 is representative of the time in which it was published and of the dataset from which it was derived. Therefore, we disagree with the premise of this comment. No change to the manuscript.

We fail to see why the reviewer is so adamant about denigrating perfectly good climatologies from the past by forcing us to use the word "outdated". Although we appreciate that some of the studies are a few years old and that new studies have been published since, such studies still offer insight into the current knowledge of hail over Europe. We have already defended our choice not to denigrate these studies. Readers will recognize that a study from 2009 is of its time, not of the present time. At what point does a study become "outdated"? Five years after publication? Ten years? Thirty years? It is unfair to the previous literature. We strongly disagree with the reviewer on this point. We already mention a more recent summary of hail studies within the same sentence (i.e. Punge and Kunz 2016). Thus, readers wanting a more recent review are already referred to this source.

Nevertheless, we can further distinguish between Tuovinen et al. (2009) and Punge and Kunz (2016). We have revised this sentence to "…Tuovinen et al. (2009), and an updated review was published by Punge and Kunz (2016)."

We have also added a new paragraph about the newer climatological studies on hail, starting at line 52.

"Climatologies of European convective storms and their impacts have been constructed using a number of datasets. For example, some studies have examined the climatology of convective storms using remote-sensed data such as lightning, radar, and satellite (e.g., Punge et al. 2017). Others have examined the environments that favor such storms, such as through reanalyses or soundings (Rädler et al. 2018; Taszarek et al. 2017, 2018, 2019) or reanalyses coupled with hailpad data (Sanchez et al. 2017)."

We hope these revisions are suitable.

**Minor Point 5:**

5. I miss a better motivation and scientific objectives of the paper. "Increasing the size of the dataset through...extending the period of analysis" is too weak when only 2 additional years are considered.

This comment is unfair. The reviewer has selectively edited this sentence to misrepresent what we actually wrote in the original submission.

"In the present article, we explore whether increasing the size of the dataset through lowering the quality-control levels of the reports and extending the period of analysis yields comparable results, increasing the generality of Púčik et al.'s (2019) results."

So, our analysis was also about adding cases through lowering the quality-control levels of the reports, not only extending the time period. These two changes resulted in an increase in the number of reports from 39,537 (Púčik et al. 2019) to 62,053 (present study), a 57% increase in the size of the dataset.

"In doing so, we also document the reporting characteristics of the database as a function of time both throughout the 20th century and within the last 20 years. In particular, we seek the possible existence of a relatively homogeneous period of time in the database that could be used as a baseline for climatologies and climate-change studies."

But, our study is about more than just increasing the size of the dataset. We also had different purposes to Púčik et al. (2019), which again were not mentioned by the reviewer.

Thus, we feel that we have clearly stated our motivation and scientific objectives, despite the manipulated and truncated quotation provided by the reviewer. No change to the manuscript.

Repeating what the manuscript already states:  The purpose of this manuscript is to create a dataset from reports with a lower quality level than used in a previous study (i.e., Púčik et al. 2019) to see if they provide similar results to higher quality-controlled reports and to determine if it is possible to identify a relatively homogeneous period of time to be used as a baseline for climate-change studies. These two pieces of text appear in the second-to-last paragraph of the introduction (right before the paragraph that describes the outline of the manuscript). This is a common location for authors to place the scientific objectives of a journal article, which is what we have done.

Moreover, we have justified the motivation and objectives of the manuscript already at the former lines 69–74, 82–84, and 114–115. We believe that the motivation and scientific objectives are clearly stated in multiple locations within the first two sections of the manuscript. We have already responded appropriately to the reviewer.

We want to emphasize again that the reviewer selectively edited our text to manipulate the interpretation of our words and to omit something we said we did, which was the principal point of doing this research project. We believe we authors have a responsibility to report this kind of unethical behavior.

**Minor 8:**

L157: "...ability to detect reports linked to the same event, and hence have removed duplicate events from the dataset". This would make no sense at all and is not the case. In the papers cited (e.g. Wilhelm et al., 2020) it is clear that a single streak is covered by several reports.

The point made here is that fewer reports have been needed for the same quantity of hail days over recent years than previously. Therefore, we are just speculating a few reasons for this. No change to the manuscript.

Just a small correction to note in this comment: The citation should be Wilhelm et al. (2021), not (2020).

We have deleted the phrase "and hence have removed duplicate events from the dataset".

**Minor 10:**

10. L188: Can you briefly describe how you converted UTC to LT?

All reports have the country of origin listed and all times are in UTC. By looking at each country on an individual basis, these were converted to LT taking daylight savings into account. No change to the manuscript.

We did not initially understand the question posed by reviewer. We now understand that there may have been a misunderstanding regarding local solar time and local legal time by time zone. In the manuscript, we converted the times from UTC to these local legal times based on each different country's time zones. We did this by searching their corresponding summer times, as the majority of reports were from April to September.

The text has been changed to the following:
"When corrected for local legal time (LT) based on each country's official time zone…."

Additionally, we have added a new Figure 7 showing the country-by-country breakdown of reports based on the time of day. There does not seem to be any particular pattern in the predominant time period for hail to form across Europe.

**Minor 11:**

11. L191-192: see comment (9); Although the diurnal cycles of Kunz et al. (2020) have a resolution of only 3 hours, there are some differences, which may be due to different study areas?

In fact, Figure 4 (local time) in the present manuscript if converted to a bar chart and Fig. 2b in Kunz et al. (2020) are quite similar. Sure, small differences will be due to different study areas and different years, but we don't see that. No change to the manuscript.

The text has been revised to "These distributions are also similar to those from Kunz et al. (2020, their Fig. 2b) who found a peak during 1500–1800 LT for hailstorms in central Europe using all quality levels from the ESWD, although small differences (e.g., relatively more hail during 1200–1500 LT in Kunz et al. (2020) compared to Fig. 5) may be due to the different study areas between these two studies."

**Minor 13:**

13. L287-289: The main reason for the high number of reports in Germany is obviously that ESSL was founded here.

Yes and no. The ESWD grew out of other data-collecting efforts such as TorDACH (tornado dataset from Germany, Austria, and Switzerland). So, although there was a focus on Germany, it was not strictly limited to the founding of ESSL. No change.

We have added new text at lines 331–334 to address the reviewers' concern.

It should be mentioned that in some countries severe weather reports are collected by other institutions, e.g. KERAUNOS in France. Moreover, crowd-sourcing via meteo apps is well known and emerging in some countries, such as the MeteoSwiss app, which has collected >100,000 reports in recent years (compared to only 266 ESWD reports). So we should not blame spotters for being less enthusiastic.

The wording as written is precise. There are two factors in play here, and our text is clear in both of those factors. Storm-spotter networks may be more or less enthusiastic about collecting reports within their own countries ("existence, size, and enthusiasm of spotter networks within each country"), and such networks may vary in how effective they are at contributing those reports to the ESWD ("variations in the ability or enthusiasm of citizens to input into the ESWD"). No change to the manuscript.

We have added this information into the manuscript about KERAUNOS and the MeteoSwiss app, etc. at lines 340–341.

It is crucial for these points to be thoroughly addressed before the manuscript can be accepted for publication. By doing so, the authors can demonstrate a commitment to improving their work and addressing the concerns of the reviewers.

Ultimately, it is the responsibility of the authors to carefully consider and respond to the concerns raised by the reviewers. The peer review process is designed to ensure the quality and rigor of scientific research, and your thorough evaluation and constructive feedback play a vital role in upholding these standards.

Again, we felt that we had addressed the reviewer's concerns in our original response. In addition, there were some comments where we disagreed with the reviewer or the reviewer did not communicate their intention clearly enough.  We apologize for any misunderstandings. We did not feel that such comments required revising the manuscript per se at that time.

Nevertheless, we have revisited those previous concerns raised by the reviewer.  We have created new figures and added new text where appropriate.  We hope these new revisions in this round are suitable.

Moreover, we were not pleased to see the reviewer selectively edit our text to change our intended meaning. This behavior is unacceptable and needs to be addressed so it does not happen in the future.

---

## Author Response (AR3)

**Response to Reviewer's Comments: Round 3**

Reviewers' comments in black.  Authors' response this round in green.

**Response to Reviewer 2**

Thank you for submitting the revised version of the manuscript. I appreciate the authors' efforts in addressing the comments raised by the reviewer. Although I still find it challenging to grasp the fundamental concept of categorizing a day as a hail day, regardless of the quantity of reports issued and their geographical distribution, I propose accepting the manuscript in its current form (but consider my comment in response to points 4 and 5).

We thank you for your comments and working with us to see this manuscript through to publication.

Regarding Major Points 4 and 5, it's noteworthy that most studies I am familiar with employ a similar definition but typically focus on single countries or specific regions. For instance, Pucik et al. (2019) determined hail days for a grid spanning 0.5° longitude and 0.5° latitude, not for all of Europe (Fig. 5 only displays diurnal cycles for Europe). The referenced review paper by Punge and Kunz (2016) discusses hail days for specific regions, not for the entire continent. Additionally, data from insurance companies, with which I have worked extensively, are usually spatially resolved. I recommend adjusting the statement in question to reflect these nuances, while also citing studies that specifically analyze thunderstorm/hail days for all of Europe without further subdivision.

Our response was to the reviewer's previous comment "Furthermore, it makes no sense to define a hail day for the whole of Europe, with its wide variety of local climates."  We argue that *it does make sense* in some situations.  The point in our response was to mention that there were studies that considered hail across all of Europe and the reasons why such studies may want to do that.  That is all we are responding to.

Our response does not negate the fact that these studies may also have investigated variability on smaller spatial scales, and our response should not be taken to imply otherwise.

We believe that the statement we wrote adequately reflects the original statement.

Punge and Kunz (2016): "Insurance records do not distinguish among different hailstorms, but register only hail damage that occurred on a whole day within a region."

Hulton and Schultz (2024): "Punge and Kunz (2016) wrote that hail days are also aligned with information that the insurance industry uses, as their portfolios cover regions larger than countries and hailstorm outbreaks may cover more than one country."

Thus, we see no reason for further revision.

I want to express my gratitude for the inclusion of the new figures, which significantly enhance the manuscript's scientific value.

You are welcome.

Concerning Minor Point 3, I realize my previous comment may have been unclear. What I

intended to convey is that the review by Tuovinen et al. (2009) predominantly cites studies from an era predating data from remote sensing instruments or reports collected through crowd sourcing or community contributions (such as ESWD). Some sources of severe hail day information rely on damage to agricultural crops or property, both of which have undergone substantial changes in the past decades. For example, Schwind's study in 1957, based on crop insurance data, identified the highest hail frequency in the Rhine valley in Germany. However, more recent studies indicate that this region is not frequently affected by hail. This discrepancy can be attributed to the fact that, at that time, tobacco was a major crop in the area, and it is highly susceptible to sleet, graupel, and small hail.

Interesting.  Thank you for providing this perspective.

I am satisfied with the newly added paragraph.

We are pleased.

Regarding Minor Point 5, I find it difficult to comprehend the authors' frustration with this issue and prefer not to delve further into it. My only request was for the objectives to be articulated more clearly in the introduction.

Similarly, we fail to understand your perspective about what is unclear about what we have written. Thus, we will have to agree to disagree on this point.

For Minor Points 8, 10, 11, and 13, I concur with the proposed changes.

Thank you.

Finally, in response to the last comment, I am perplexed as to where I should selectively edit the text to change the authors' intended meaning. The comment seems unusual, and the reaction appears disproportionate. However, I choose not to elaborate further on this matter.

Similarly, we fail to understand why you would have selectively edited our words to omit key concepts in our response. From our perspective, that is what appeared to happen.

**Response to Reviewer 4**

• Lines 124-128.
o I believe the content here addresses the previously raised point but the wording could be improved. I believe it is acceptable to consider hail days across the whole of Europe and that introducing additional thresholds only shifts the debate to which threshold to use.

Thank you for your agreement that the concept of hail days across Europe makes sense.

• Generally relating to counting hail days
o In recent years there are so many hail days that it ceases to become a useful measure of convective activity. I do not think it is an invalid metric to study but do wonder whether some additional criteria of severe hail days could be added.

We are unclear what the reviewer is asking for here with "some additional criteria". Nevertheless, we have tried to address in our response below why hail days were a suitable choice.

We also consider the number of hail reports and highlight these by year (Figures 1 and 2) and by month (Figure 3) to highlight that there has been a change in how hail is reported and how this impacts the number of hail days recorded. Figure 2 in particular shows that it is only in the past 20 years or so that there seem to be a more stable relationship between hail reports and hail days, which could suggest a more stable time period within which most hail events are being recorded. This stable time period can also be seen in Figure 9 which shows little variation in the percentage distribution of each hail bin size from 2000 to 2020.

We also break down the number of hail reports by month by country (Figure 4), which shows that the warm season is more likely to be affected by these events, with little variation between countries across the continent.

Figure 14 also shows how the hail sizes are distributed by country, showing which areas are at most risk of seeing the largest hail. Hail days are not the only metric used in this study.

We hope this response helps convey that no additional criteria are needed.

• Additional Comments
o I think the paper falls slightly short in its aim of evaluating the ESWD in any critical manner. The analysis of the observations is useful and may hopefully form a useful reference to anyone using the ESWD but the conclusions section could be embellished to truly evaluate the usefulness of the EWSD and its shortcomings.

Again, we are not entirely sure what else we could say to address the reviewer's concern that we have not "truly evaluated the usefulness of the EWSD". In some sense, that usefulness will be dependent upon the user's interest. Nevertheless, we hope our response to the comment below does help us go forward.

o I would like to see some break down of the stable time period discussion by region to assess whether some areas are now reaching a saturation of reporting. It may also be worth considering at what point is the reporting sufficient that all major hail swathes are captured and additional reporting is only increasing the density or reports within swathes. This may be beyond the scope of this work but is essential for assessing the extent to which the ESWD can be used as a hail climatology. Section 3 does comment on this briefly.

We have added an annual large-hail day breakdown by country for the countries with 100+ reports for the period 2000–2020 (Fig. 14 a, b, c, and d). Here we demonstrate that the countries with the most annual-hail days have started to show a relatively consistent quantity of hail days over the past few years, which could suggest that all major hail swaths are being captured in the dataset. We also show that for those countries with fewer large-hail days, the quantity observed has increased over the past few years, which suggests and increasing in reporting in these regions, and hence expands the usefulness of the database by encompassing more events. However, we do also highlight that there remains much annual variability in hail-days per country, which could be due to meteorological and climatic factors, or may be more to do with reporting.

We have also added the following sentence to the conclusion: "When considering the annual number of large-hail days per country, there does appear to be an overall increase in the quantity observed for the countries which previously reported fewer hail-days, while those which observed greater numbers throughout this period seem to be stabilizing." (L525)

Yes, we agree that the point of sufficient hail reporting is beyond this particular work, but is an excellent future research project.  Thank you again for your kind words and support.

---

## Author Response (AR4)

**Response to Reviewer's Comments: Round 4**

Reviewers' comments in black. Authors' response this round in green.

Dear authors, thanks for your the revised version of the manuscript, which I am largely happy with. There is one exception, concerning your reply to a comment from reviewer 2 regarding Major Points 4 and 5 of the initial review. I think we need to find an appropriate solution for this, and basically it is centered around the following statement from Punge and Kunz (2016)

"Insurance records do not distinguish among different hailstorms, but register only hail damage that occurred on a whole day within a region."

... and the choice of the authors to define hail days for a larger area. I suggest to combine the above statement with the new sentence

"Punge and Kunz (2016) wrote that hail days are also aligned with information that the insurance industry uses, as their portfolios cover regions larger than countries and hailstorm outbreaks may cover more than one country."

A suggest could be something similar to:

Punge and Kunz (2016) wrote that Insurance records do not distinguish among different hailstorms, but register only hail damage that occurred on a whole day within a region. However, as insurance portfolios cover regions larger than countries and hailstorm outbreaks may cover more than one country, it is meaningful to combine this information to hail days for (large parts of) Europe

Looking forward to the revised version of the manuscript

We thank you for your comments and working with us to see this manuscript through to publication.

We appreciate the point highlighted here and have modified/added the following sentence for clarity:

L126-130: "For example, Punge and Kunz (2016) wrote that the insurance industry measures hail damage per region per day instead of measuring damage per individual hailstorm. Therefore, a pan-European overview of hail days may be of use given that these insurance portfolios cover large parts of Europe, often including data from multiple countries. However, an awareness of the spatial distribution of these reports is necessary to identify the most at-risk regions."

We hope this addresses the concern over using hail days as a metric for Europe as a whole, while acknowledging that an understanding of the spatial distribution is necessary to make identify the most at risk areas.